# Model Tensor Planning

**An T. Le**[1,7]**, Khai Nguyen**[7]**, Minh Nhat Vu**[5,6]**, João Carvalho**[1]**, Jan Peters**[1,2,3,4]

*{an, joao, jan}@robot-learning.de*

[1] *Intelligent Autonomous Systems, Department of Computer Science, Technical University of Darmstadt, Germany*

[2] *Systems AI for Robot Learning, German Research Center for AI (DFKI), Germany*

[3] *Hessian Center for Artificial Intelligence (hessian.AI), Germany*

[4] *Centre for Cognitive Science, Technical University of Darmstadt, Germany*

[5] *Automation & Control Institute, TU Wien, Austria*

[6] *Austrian Institute of Technology (AIT), Vienna, Austria*

[7] *VinRobotics and VinUniversity, Vietnam*

**Reviewed on OpenReview:** *https://openreview.net/forum?id=fk1ZZdXCE3*

## Abstract

Sampling-based model predictive control (MPC) offers strong performance in nonlinear and contact-rich robotic tasks, yet often suffers from poor exploration due to locally greedy sampling schemes. We propose *Model Tensor Planning* (MTP), a novel sampling-based MPC framework that introduces high-entropy control trajectory generation through structured tensor sampling. By sampling over randomized multipartite graphs and interpolating control trajectories with B-splines and Akima splines, MTP ensures smooth and globally diverse control candidates. We further propose a simple $\beta$-mixing strategy that blends local exploitative and global exploratory samples within the modified Cross-Entropy Method (CEM) update, balancing control refinement and exploration. Theoretically, we show that MTP achieves asymptotic path coverage and maximum entropy in the control trajectory space in the limit of infinite tensor depth and width.

Our implementation is fully vectorized using JAX and compatible with MuJoCo XLA, supporting *Just-in-time* (JIT) compilation and batched rollouts for real-time control with online domain randomization. Through experiments on various challenging robotic tasks, ranging from dexterous in-hand manipulation to humanoid locomotion, we demonstrate that MTP outperforms standard MPC and evolutionary strategy baselines in task success and control robustness. Design and sensitivity ablations confirm the effectiveness of MTP's tensor sampling structure, spline interpolation choices, and mixing strategy. Altogether, MTP offers a scalable framework for robust exploration in model-based planning and control.

## 1 Introduction

Sampling-based Model Predictive Control (MPC) (Mayne, 2014; Lorenzen et al., 2017; Williams et al., 2017) has emerged as a powerful framework for controlling nonlinear and contact-rich systems. Unlike gradient-based or linearization approaches, sampling-based MPC is model-agnostic and does not require differentiable dynamics, making it well-suited for high-dimensional, complex systems such as legged robots (Alvarez-Padilla et al., 2024) and dexterous manipulators (Li et al., 2024). Moreover, its inherent parallelism enables efficient deployment on modern hardware (e.g., GPUs), allowing for high-throughput simulation and online domain randomization in real-time control pipelines (Pezzato et al., 2025).

Despite these advantages, a fundamental limitation remains: sampling-based MPC is typically local in its search behavior. Most methods perturb a nominal trajectory or refine the current best samples, which makes them susceptible to local minima and unable to consistently discover globally optimal solutions (Xue et al., 2024). While the Cross-Entropy Method (CEM) (De Boer et al., 2005) has shown promise in high-dimensional control and sparse-reward settings (Pinneri et al., 2021), it still suffers from mode collapse

when sampling locally (Zhang et al., 2022), leading to suboptimal behaviors. The curse of dimensionality exacerbates this issue, as the number of samples required to explore control spaces grows exponentially with the planning horizon and control dimension, posing a bottleneck if compute or memory is limited. These challenges motivate the need for a more effective, high-entropy sampling mechanism for control generation.

Evolutionary Strategies (ES) (Hansen, 2016; Wierstra et al., 2014; Salimans et al., 2017) have also been applied in sampling-based MPC settings to improve sampling exploration. While they improve over purely local strategies in some tasks, our experiments (cf. Section 4.1) reveal that ES still fails to systematically explore multimodal control landscapes, often yielding inconsistent performance on tasks requiring long-term coordinated actions.

In this work, we introduce *Model Tensor Planning* (MTP), a novel sampling-based MPC framework that enables globally exploratory control generation through structured tensor sampling. MTP reformulates control sampling as tensor operations over randomized multipartite graphs, enabling efficient generation of diverse control sequences with high entropy. To balance exploration and exploitation, we propose a simple yet effective $\beta$-mixing mechanism that combines globally exploratory samples with locally exploitative refinements. We also provide a theoretical analysis under bounded-variation assumptions, showing that our sampling scheme achieves asymptotic path coverage, approximating maximum entropy in trajectory space.

MTP is designed with real-time applicability with matrix-based formulation, which is compatible with *Just-in-time* (JIT) compilation and vectorized mapping (e.g., via JAX `vmap` (Bradbury et al., 2018)), allowing high-throughput sampling, batch rollout evaluation, and online domain randomization on modern simulators. Our main contributions are as follows:

- We propose *tensor sampling*, a novel structured sampling strategy for control generation, and provide theoretical justification via asymptotic path coverage.

- We introduce a simple $\beta$-mixing mechanism that effectively balances exploration and exploitation by blending high-entropy and local samples within the modified CEM update rule.

- We demonstrate that MTP is highly compatible with modern vectorized simulators, enabling efficient batch rollout evaluation and robust real-time control in high-dimensional, contact-rich environments.

## 2 Preliminary

**Notations and Assumptions.** We consider the problem of sampling-based MPC. Given a dynamics model $\dot{\boldsymbol{x}} = f(\boldsymbol{x}, \boldsymbol{u})$, we consider the path sampling problem in the control space $\mathbb{U} \subset \mathbb{R}^n$ with the control $\boldsymbol{u} \in \mathbb{U}$ having $n$-dimensions at the current system state $\boldsymbol{x} \in \mathbb{X} \subset \mathbb{R}^d$. Typically, a batch of control trajectories is sampled, rolled out through the dynamics model, and evaluated using a cost function. Let a control path be $u : [0, 1] \to \boldsymbol{u}$, $\boldsymbol{u}(t) \in \mathbb{U}$, we can define the path arc length as

$$\mathrm{TV}(u) = \sup_{M \in \mathbb{N}^+, 0 = t_1, \dots, t_M = 1} \sum_{i=1}^{M-1} \|\boldsymbol{u}(t_{i+1}) - \boldsymbol{u}(t_i)\|. \tag{1}$$

We define $\mathbb{F}$ as the set of all control paths that are uniformly continuous with bounded variation $TV(u) < \infty$, $u \in \mathbb{F}$. This assumption is common in many control settings, where the control trajectories are bounded in time and control space (i.e., both time and control spaces are compact). Throughout this paper, we narrate the preliminary and the tensor sampling method in matrix definitions, discretizing continuous paths with equal time intervals.

### 2.1 Cross-Entropy Method for Sampling-based MPC

Consider a discretized dynamical system $\boldsymbol{x}_{t+1} = f(\boldsymbol{x}_t, \boldsymbol{u}_t)$, $t = 0, \dots, T-1$ with horizon $T$, where $\boldsymbol{x}_t \in \mathbb{R}^d$, $\boldsymbol{u}_t \in \mathbb{R}^n$ denotes the state the control at time step $t$. Given the state cost function $c(\boldsymbol{x}, \boldsymbol{u})$ and the terminal cost $c_T(\boldsymbol{x})$, the objective is to minimize a cumulative cost function

$$J(\boldsymbol{\tau}, \boldsymbol{U}) = \sum_{t=0}^{T-1} c(\boldsymbol{x}_t, \boldsymbol{u}_t) + c_T(\boldsymbol{x}_T), \text{ s.t. } \boldsymbol{x}_{t+1} = f(\boldsymbol{x}_t, \boldsymbol{u}_t) \tag{2}$$

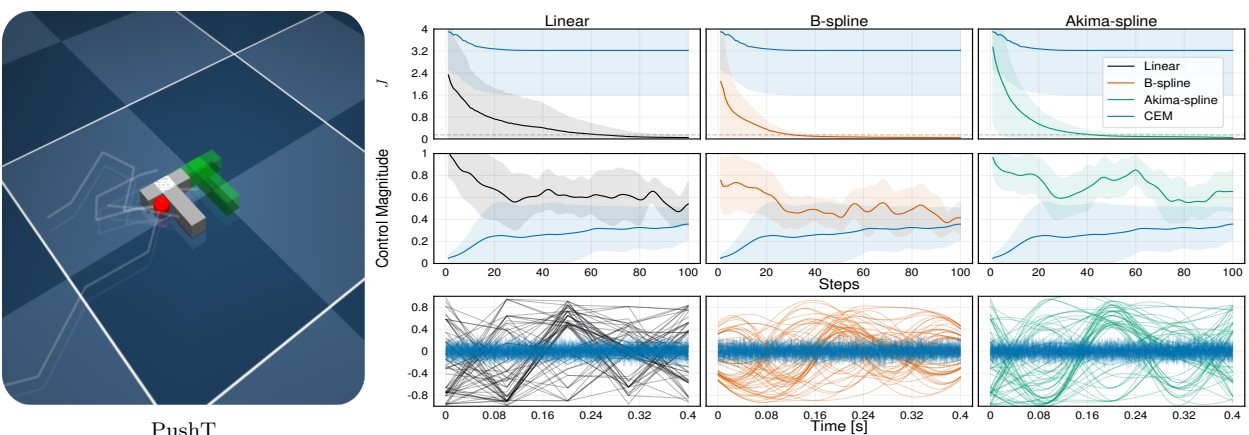

Figure 1: Comparison of MTP interpolation methods versus CEM on `PushT` environment. The cost of `PushT` is the sum of the position and orientation error to the green target (without the guiding contact cost), and the initial T pose is randomized. The first row depicts the cost convergence over 10 seeds. In most seeds, CEM struggles to push the object due to the lack of exploration (e.g., mode collapsing), while MTP variants always find the correct contact point to achieve the task. Note that the control magnitude of MTP is high due to the global explorative samples (see second & third rows), compared to the white noise samples (blue). B-spline helps regulate the control magnitude due to its barycentric weightings, while retaining exploration behaviors. The last row illustrates the control trajectories between 64 tensor samples and 64 white noise trajectories. Experiment videos are publicly available at `https://sites.google.com/view/tensor-sampling/`.

where the state trajectory $\boldsymbol{\tau} = [\boldsymbol{x}_0, \ldots, \boldsymbol{x}_T] \in \mathbb{R}^{(T+1) \times d}$, and control trajectory $\boldsymbol{U} = [\boldsymbol{u}_0, \ldots, \boldsymbol{u}_{T-1}] \in \mathbb{R}^{T \times n}$ are the dynamics rollout, $c(\boldsymbol{x}_t, \boldsymbol{u}_t)$ is the immediate cost at each time step, and $c_T(\boldsymbol{x}_T)$ is the terminal cost.

CEM optimizes the control sequence $\boldsymbol{U}$ iteratively by approximating the optimal control distribution using a parametric probability distribution, typically Gaussian. Initially, the control inputs are sampled from an initial distribution parameterized by mean $\boldsymbol{\mu} \in \mathbb{R}^{T \times n}$ and standard deviation $\boldsymbol{\sigma} \in \mathbb{R}^{T \times n}$. At each iteration, CEM performs the following steps iteratively:

**Sampling.** Draw $B$ candidate control sequences from the current Gaussian distribution:

$$\boldsymbol{U}^{(k)} \sim \mathcal{N}(\boldsymbol{\mu}, \mathrm{diag}(\boldsymbol{\sigma}^2)), \quad k = 1, \ldots, B. \tag{3}$$

**Evaluation.** Rollout $\boldsymbol{\tau}^{(k)}$ from the dynamics model and compute the cost $J(\boldsymbol{\tau}^{(k)}, \boldsymbol{U}^{(k)})$ for each sampled control sequence by simulating the system dynamics.

**Elite Selection.** Choose the top-$E \leq B$ elite candidates of control sequences that have the lowest cost, forming an elite set $\mathcal{E} = \{\boldsymbol{U}^{(k)}\}_{k=1}^{E}$.

**Distribution Update.** Update the parameters $(\boldsymbol{\mu}, \boldsymbol{\sigma})$ based on elite samples:

$$\boldsymbol{\mu}_{\mathrm{new}} = \frac{1}{E} \sum_{\boldsymbol{U} \in \mathcal{E}} \boldsymbol{U}, \quad \boldsymbol{\sigma}_{\mathrm{new}}^2 = \frac{1}{E-1} \sum_{\boldsymbol{U} \in \mathcal{E}} (\boldsymbol{U} - \boldsymbol{\mu}_{\mathrm{new}})^2. \tag{4}$$

An exponential smoothing update rule can be used for stability:

$$\boldsymbol{\mu} \leftarrow \alpha \boldsymbol{\mu} + (1 - \alpha) \boldsymbol{\mu}_{\mathrm{new}}, \quad \boldsymbol{\sigma} \leftarrow \alpha \boldsymbol{\sigma} + (1 - \alpha) \boldsymbol{\sigma}_{\mathrm{new}}, \tag{5}$$

where $\alpha \in [0, 1)$ is a smoothing factor.

**Termination Criterion.** The iterative process continues until a convergence criterion is met or a maximum number of iterations is reached. The optimal control sequence is approximated by the final mean $\boldsymbol{\mu}$ of the Gaussian distribution.

## 2.2 Spline-based Controls

Splines provide a powerful representation for trajectory generation in MPC due to their flexibility, continuity properties, and ease of parameterization (Jankowski et al., 2023; Carvalho et al., 2025). In this work, we focus on spline-parametrization of control trajectories. Spline-based trajectories ensure smooth and feasible control inputs that satisfy constraints and objectives inherent to MPC frameworks.

A spline is defined as a piecewise polynomial function $\boldsymbol{u}(t) : [0, T] \to \mathbb{U}$, which is polynomial within intervals divided by knots $t_1, \ldots, t_M$, with continuity conditions enforced at these knots. In particular,

- **Knots.** A knot $t_i \in [0, T]$ is a time point where polynomial pieces join. We have a non-decreasing sequence of knots $0 = t_1 \leq \ldots \leq t_M = T$, which partition the time interval $[0, T]$ into pieces $[t_i, t_{i+1}]$ so that the path is polynomial in each piece. We may often have double or triple knots, meaning that several consecutive knots $t_i = t_{i+1}$ are equal, especially at the beginning and end, as this can ensure boundary conditions for zero higher-order derivatives.

- **Waypoints.** A waypoint $\boldsymbol{u}(t) \in \mathbb{U}$ is a point on the path, typically corresponding to $\boldsymbol{u}(t_i)$.

- **Control Points.** A set of control points $\mathcal{Z} = \{\boldsymbol{z}_i | \boldsymbol{z}_i \in \mathbb{R}^n\}_{i=1}^K$ parametrizes the spline via basis functions.

**B-Spline Parameterization.** In B-splines (de Boor, 1973), the path $\boldsymbol{u}$ is expressed as a linear combination of control points $\boldsymbol{z}_i \in \mathcal{Z}$

$$\boldsymbol{u}(t) = \sum_{i=1}^K B_{i,p}(t)\boldsymbol{z}_i, \quad \text{s.t.} \quad \sum_{i=1}^K B_{i,p}(t) = 1, \tag{6}$$

where $B_{i,p} : \mathbb{R} \to \mathbb{R}$ maps the time $t$ to the weighting of the $i^{\text{th}}$ control point, depicting the $i^{\text{th}}$ control point weight for blending (i.e., as with a probability distribution over $i$). Hence, the control waypoint $\boldsymbol{u}(t)$ is always in the convex hull of control points. The B-spline functions $B_{i,p}(t)$ are fully specified by a non-decreasing series of time knots $0 = t_1 \leq \ldots \leq t_M = T$ and the integer polynomial degree $p \in \{0, 1, \ldots\}$ by

$$B_{i,0}(t) = [t_i \leq t \leq t_{i+1}], \quad \text{for } 1 \leq i \leq M - 1,$$
$$B_{i,p}(t) = \frac{t - t_i}{t_{i+p} - t_i} B_{i,p-1}(t) + \frac{t_{i+p+1} - t}{t_{i+p+1} - t_{i+1}} B_{i+1,p-1}(t), \quad \text{for } 1 \leq i \leq M - p - 1. \tag{7}$$

$B_{i,0}$ are binary indicators of $t \in [t_i, t_{i+1}]$ with $1 \leq i \leq M - 1$. The 1st-degree B-spline functions $B_{i,1}$ have support in $t \in [t_i, t_{i+2}]$ with $1 \leq i \leq M - 2$, such that $\sum_{i=1}^{M-2} B_{i,1}(t) = 1$ holds. In general, degree $p$ B-spline functions $B_{i,p}$ have support in $t \in [t_i, t_{i+p+1}]$ with $1 \leq i \leq M - p - 1$. We need $K = M - p - 1$ control points $\boldsymbol{z}_{1:K}$, which ensures the normalization property $\sum_{i=1}^K B_{i,p}(t) = 1$ for every degree.

**Akima-Spline Parameterization.** The Akima spline (Akima, 1974) is a piecewise cubic interpolation method that exhibits $C^1$ smoothness by using local points to construct the spline, avoiding oscillations or overshooting in other interpolation methods, such as B-splines. In other words, an Akima spline is a piecewise cubic spline constructed to pass through control points with $C^1$ smoothness. Given the control point set $\mathcal{Z}$ with $K = M$, the Akima spline constructs a piecewise cubic polynomial $u(t)$ for each interval $[t_i, t_{i+1}]$

$$\boldsymbol{u}_i(t) = \boldsymbol{d}_i(t - t_i)^3 + \boldsymbol{c}_i(t - t_i)^2 + \boldsymbol{b}_i(t - t_i) + \boldsymbol{a}_i, \tag{8}$$

where the coefficients $\boldsymbol{a}_i, \boldsymbol{b}_i, \boldsymbol{c}_i, \boldsymbol{d}_i \in \mathbb{U}$ are determined from the conditions of smoothness and interpolation. Let $\boldsymbol{m}_i = (\boldsymbol{z}_{i+1} - \boldsymbol{z}_i)/(t_{i+1} - t_i)$, the spline slope is computed as

$$\boldsymbol{s}_i = \frac{|\boldsymbol{m}_{i+1} - \boldsymbol{m}_i|\boldsymbol{m}_{i-1} + |\boldsymbol{m}_{i-1} - \boldsymbol{m}_{i-2}|\boldsymbol{m}_i}{|\boldsymbol{m}_{i+1} - \boldsymbol{m}_i| + |\boldsymbol{m}_{i-1} - \boldsymbol{m}_{i-2}|}. \tag{9}$$

The spline slopes for the first two points at both ends are $\boldsymbol{s}_1 = \boldsymbol{m}_1, \boldsymbol{s}_2 = (\boldsymbol{m}_1 + \boldsymbol{m}_2)/2, \boldsymbol{s}_{M-1} = (\boldsymbol{m}_{M-1} + \boldsymbol{m}_{M-2})/2, \boldsymbol{s}_M = \boldsymbol{m}_{M-1}$. Then, the polynomial coefficients are uniquely defined

$$\boldsymbol{a}_i = \boldsymbol{u}_i, \quad \boldsymbol{b}_i = \boldsymbol{s}_i, \quad \boldsymbol{c}_i = (3\boldsymbol{m}_i - 2\boldsymbol{s}_i - \boldsymbol{s}_{i+1})/(t_{i+1} - t_i), \quad \boldsymbol{d}_i = (\boldsymbol{s}_i + \boldsymbol{s}_{i+1} - 2\boldsymbol{m}_i)/(t_{i+1} - t_i)^2. \tag{10}$$

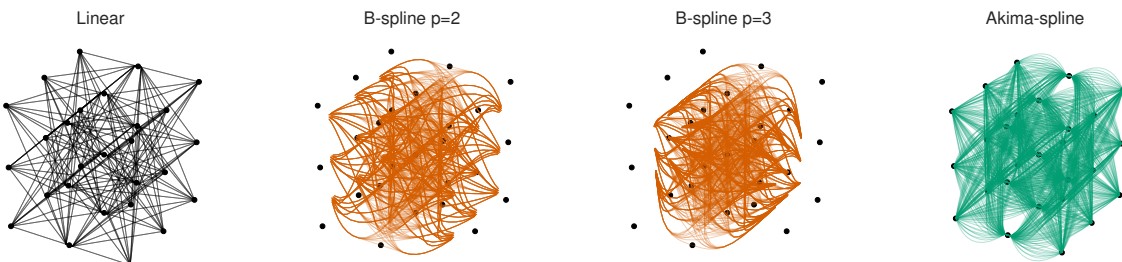

Figure 2: Illustration of different tensor interpolations on an evenly spaced graph with $M = 3, N = 9$. With a higher B-spline degree, the control trajectories exhibit more smoothness and conservative behavior, while Akima-spline aggressively and smoothly tracks control-waypoints. Note that we do not consider boundary conditions for B-spline interpolation in tensor sampling.

**Motivation.** Spline representation provides several benefits. (i) It ensures smooth control trajectories (Watson & Peters, 2023) with guaranteed continuity of positions, velocities, and accelerations under mild assumptions of the dynamics. (ii) Spline simplifies complex trajectories through a few control points and efficiently incorporates constraints and boundary conditions, enabling efficient learning and planning by dimension reduction (Carvalho et al., 2025). (iii) It enables easy numerical optimization thanks to differentiable and convex representations.

## 3 Method

We first propose *tensor sampling* – a batch control path sampler having high explorative behavior, and investigate its path coverage property over the compact control space. Then, we incorporate the tensor sampling with the modified CEM update rule, balancing exploration and exploitation in cost evaluation, forming an overall vectorized sampling-based MPC algorithm.

### 3.1 Tensor Sampling

Inspired by (Le et al., 2025), we utilize the random multipartite graph as a tensor discretization structure to approximate global path sampling.

**Definition 1** (Random Multipartite Graph Control Discretization)**.** *Consider a graph $\mathcal{G}(M, N) = (V, W)$ on control space $\mathbb{U}$, the node set $V = \{L_i\}_{i=1}^M$ is a set of $M$ layers. Each layer $L_i = \{\boldsymbol{u}_j \in \mathbb{U} \mid \boldsymbol{u}_j \sim Uniform(\mathbb{U})\}_{j=1}^N$ contains $N$ control-waypoints sampled from a uniform distribution over control space (i.e., bounded by control limits). The edge set $W$ is defined by the union of (forward) pair-wise connections between layers*

$$W = \{(\boldsymbol{u}_i, \boldsymbol{u}_{i+1}) \mid \forall \boldsymbol{u}_i \in L_i, \ \boldsymbol{u}_{i+1} \in L_{i+1}, 1 \le i < M\},$$

*leading to a complete $M$-partite directed graph.*

**Sampling from $\mathcal{G}(M, N)$.** The graph nodes are represented as the control-waypoint tensor for all layers $\boldsymbol{Z} \in \mathbb{R}^{M \times N \times n}$, within the control limits. To sample a batch of $B \in \mathbb{N}^+$ control paths with a horizon $T$, we subsample with replacement $\boldsymbol{C} \in \mathbb{R}^{B \times M \times n}$ from the set of all combinatorial paths in $\mathcal{G}$ (cf. Algorithm 1) and further interpolate $\boldsymbol{C}$ into control trajectories $\boldsymbol{U} \in \mathbb{R}^{B \times T \times n}$ with different smooth structures, e.g., using Eq. (6) or Eq. (8). Sampling with replacement is cheap $\mathcal{O}(MN)$, while sampling without replacement is $\mathcal{O}(N^M)$. Sampling without replacement adds overheads due to re-indexing or tree-traversing to get batch of sequence indices (depending on the low-level implementation of sampling) but offers better diversity. In practice, we use sampling with replacement, which does not really affect diversity (see last row in Fig. 1), is faster and scales well with JAX vectorized operations. To see this, we have $N^M$ combinatorial paths in $\mathcal{G}(M, N)$, each path in $\mathcal{G}(M, N)$ has uniform $1/N^M$ mass, and the probability of sampling the same paths is small.

**Control Path Interpolation**. Straight-line interpolation can be realized straightforwardly by simply probing an $H = \lfloor T/M \rfloor$ number of equidistant points between layers, forming the linear coverage trajectories

$U \in \mathbb{R}^{B \times T \times n}$. However, there exist discontinuities at control waypoints using linear interpolation. Hence, we motivate spline tensor interpolations for sampling smooth control paths (cf. Fig. 2).

**Definition 2** (B-spline Control Trajectories). *Given two time sequences $t_i = i/(M+p+1)$, $i \in \{0, \ldots, M+p\}$ and $t_j = j/T$, $j \in \{0, \ldots, T-1\}$, the B-Spline matrix $\boldsymbol{B}_p \in \mathbb{R}^{M \times T}$ can be constructed recursive following Eq. (7), with index $i, j$ corresponding to the element $\boldsymbol{B}_{i,p}(t_j)$. Then, the control trajectories can be interpolated by performing* `einsum` *on the $M$ dimension of $\boldsymbol{B}_p \in \mathbb{R}^{M \times T}$, $\boldsymbol{C} \in \mathbb{R}^{B \times M \times n}$, resulting $\boldsymbol{U} \in \mathbb{R}^{B \times T \times n}$.*

B-spline control trajectories exhibit conservative behavior as they are strictly inside the control point convex hull. Alternatively, they can be forced to pass through all control points by adding a multiplicity of $p$ per knot (de Boor, 1973). However, this method wastes computation by increasing the B-spline matrix size to $Mp \times N$, and still cannot avoid the overshooting problem. Thus, we further propose the Akima-spline control interpolation.

**Definition 3** (Akima-spline Control Trajectories (Le et al., 2025)). *Given the time sequences $t_i = i/M$, $i \in \{0, \ldots, M-1\}$ representing the $M$ layer time slices and the control points $\boldsymbol{C} \in \mathbb{R}^{B \times M \times n}$, the Akima polynomial parameters $\boldsymbol{A} \in \mathbb{R}^{B \times (M-1) \times 4 \times n}$ can be computed following Eq. (10) in batch. Then, given the time sequence $t_j = j/T$, $j \in \{0, \ldots, T-1\}$, the control trajectories $\boldsymbol{U} \in \mathbb{R}^{B \times T \times n}$ are interpolated following polynomial interpolation Eq. (8).*

In the next section, we investigate whether the path distribution support of tensor sampling approximates the support of all possible paths in the control space, with linear interpolation. B-spline and Akima-spline variants are deferred for future work.

## 3.2 Path Coverage Guarantee

We analyze the path coverage property of $\mathcal{G}(M, N)$ for sampling control paths with linear interpolation. In particular, we investigate that any feasible path in the control space can be approximated arbitrarily well by a path in the random multipartite graph $\mathcal{G}(M, N)$ as the number of layers $M$ and the number of waypoints per layer $N$ approach infinity.

**Theorem 1** (Asymptotic Path Coverage). *Let $u \in \mathbb{F}$ be any control path and $\mathcal{G}(M, N)$ be a random multipartite graph with $M$ layers and $N$ uniform samples per layer (cf. Definition 1). Assuming a time sequence (i.e., knots) $0 = t_1 < t_2 < \ldots < t_M = 1$ with equal intervals, associating with layers $L_1, \ldots, L_M \in \mathcal{G}(M, N)$ respectively, then*

$$\lim_{M,N \to \infty} \min_{g \in \mathcal{G}(M,N)} \|u - g\|_\infty = 0.$$

In intuition, Theorem 1 states the support of path distribution $\mathcal{G}(M, N)$ approximates $\mathbb{F}$ and converges to $\mathbb{F}$ as $M, N \to \infty$. Thus, sampling paths from $\mathcal{G}(M, N)$ provides a tensorized mechanism to efficiently sample any possible path from $\mathbb{F}$, which allows vectorized sampling.

**Remark.** *As $M, N \to \infty$, for any control path $g \in \mathbb{F}$, then $g \in \mathcal{G}(M, N)$. Hence, $\mathcal{G}(M, N)$ represents all homotopy classes in the limit $M, N \to \infty$, and sampling paths from $\mathcal{G}(M, N)$ approximate sampling from all possible paths.*

To quantify the exploration level of tensor sampling, one standard way is to investigate its path distribution entropy. In intuition, when $M, N \to \infty$, tensor sampling entropy also approaches infinity due to uniform sampling per layer, which is further discussed in Appendix A.2. Note that this theory investigation serves as a guiding principle, while practical settings trade off entropy with success rate and runtime feasibility.

---

**Algorithm 1:** Sampling Paths From $\mathcal{G}(M, N)$

**Input:** Control waypoints $\boldsymbol{Z} \in \mathbb{R}^{M \times N \times n}$, number of paths $B$
**Output:** Sampled control-waypoints $\boldsymbol{C} \in \mathbb{R}^{B \times M \times n}$

1   $\boldsymbol{I} \leftarrow$ `randint`$((B, M), 1, N)$. // `batch sample with replacement from 1,...,N with shape (B,M)`
2   $\boldsymbol{C} \leftarrow$ `parse_index`$(\boldsymbol{Z}, \boldsymbol{I})$. // `extract waypoints from sampled indices into` $C \in \mathbb{R}^{B \times M \times n}$

---

**Practical Settings.** We typically only set $M < T$. In principle, increasing $N$ should enable finer-grained exploration over the trajectory space. However, we observe diminishing returns when $N$ increases while keeping the total number of sampled trajectories $B$ fixed (cf. Fig. 8). Intuitively, the underlying graph becomes denser, and the number of explored paths remains constant, resulting in only marginal performance improvements. Therefore, we recommend choosing $N$ proportionally to $B$ and within the bounds of available GPU memory, to maintain computational efficiency without oversampling from a small sample size.

### 3.3  Algorithm

Here, we present the overall algorithm combining tensor and local sampling with smooth structure options in Algorithm 2. We propose a simple mixing mechanism with $\beta \in [0, 1]$ by concatenating explorative and exploitative samples, forming a control trajectory tensor $\boldsymbol{U}$ (Line 4-8). We include the current nominal control for system stability at the fixed-point states (e.g., for tracking tasks) (Howell et al., 2022). Using simulators that allow for parallel runs (Todorov et al., 2012; Makoviychuk et al., 2021), rollout and cost evaluation can be efficiently vectorized (Line 9).

To tame the noise induced by tensor sampling, we modify the CEM update with `softmax` weighting on the elite set, for computing the new weighted control means and covariances similar to the MPPI update rule (Williams et al., 2017). We observe that this greatly smoothens the update over timesteps (Line 11-13) (cf. Fig. 9). Finally, similar to Howell et al. (2022), we send the first control of the best candidate, since this control trajectory is evaluated in the simulator rather than the updated mean $\boldsymbol{\mu}$. Notice that we have fixed tensor shapes based on hyperparameters, for all subroutines of sampling, rolling out, and control distribution updates. Algorithm 2 can be JAX `jit` and `vmap` over a number of $R$ model perturbations $\{f_j(\boldsymbol{x}, \boldsymbol{u})\}_{j=1}^{R}$, for efficient online domain randomization, while maintaining real-time control (cf. Appendix A.5).

---

**Algorithm 2:** Model Tensor Planning

  **Input:** Model $f(\boldsymbol{x}, \boldsymbol{u})$, graph params $M, N$, num samples $B$, mixing rate $\beta \in [0, 1]$, planning
  horizon $T$. CEM params $\alpha \in [0, 1), \lambda > 0, \sigma_m > 0, E$, which are smooth and temperature
  scalar, minimum variance, elite number, respectively.

**1** Choose interpolation type **Linear**, or **B-spline** (Definition 2), or **Akima** (Definition 3).

**2** Init the nominal control $\boldsymbol{\mu} \in \mathbb{R}^{T \times n}$ and variance $\mathrm{diag}(\boldsymbol{\sigma}^2), \boldsymbol{\sigma} \in \mathbb{R}^{T \times n}$.

**3 while** *Task is not complete* **do**

      // `tensor sampling`

**4**     Uniformly sample $\boldsymbol{Q} \in \mathbb{R}^{M \times N \times d}$ on control space $\mathbb{U}$.

**5**     Sample control waypoints $\boldsymbol{C} \in \mathbb{R}^{P \times M \times n}$ with $P = \lfloor \beta B \rfloor$, using Algorithm 1.

**6**     Interpolate $\boldsymbol{C}$ using with chosen interpolation method into control trajectories $\boldsymbol{U}_{\mathcal{G}} \in \mathbb{R}^{P \times T \times n}$.

      // `local sampling`

**7**     Sample $B - P - 1$ local trajectories $\boldsymbol{U}_{\mathrm{Local}} \sim \mathcal{N}(\boldsymbol{\mu}, \mathrm{diag}(\boldsymbol{\sigma}^2))$.

**8**     Stack $\boldsymbol{U}_{\mathrm{Local}}, \boldsymbol{U}_{\mathcal{G}}, \boldsymbol{\mu}$ into $\boldsymbol{U} \in \mathbb{R}^{B \times T \times n}$

      // `Update routine using vectorized simulator`

**9**     Batch rollout $\boldsymbol{X} \in \mathbb{R}^{B \times T \times d}$ from model $f(\boldsymbol{x}, \boldsymbol{u})$, and evaluate cost matrix $\boldsymbol{S}(\boldsymbol{X}, \boldsymbol{U}) \in \mathbb{R}^{B \times T}$.

**10**     Sum cost $\boldsymbol{s} = \sum_t \boldsymbol{S}_{:,t} \in \mathbb{R}^B$ and sort top-$E$ elite candidate indices $\boldsymbol{i}_E$.

**11**     Select candidate $\boldsymbol{U} \leftarrow \texttt{parse\_index}(\boldsymbol{U}, \boldsymbol{i}_E)$ and compute candidate weights $\boldsymbol{w} = \dfrac{\exp\left(-\frac{1}{\lambda} \boldsymbol{s}[\boldsymbol{i}_E]\right)}{\sum \exp\left(-\frac{1}{\lambda} \boldsymbol{s}[\boldsymbol{i}_E]\right)}$.

**12**     Compute new mean $\boldsymbol{\mu}' = \boldsymbol{w}\boldsymbol{U}$ and variance $\boldsymbol{\sigma}' = \max(\boldsymbol{w} \sum_{\boldsymbol{u} \in \boldsymbol{U}} (\boldsymbol{u} - \boldsymbol{\mu}')^2, \sigma_m)$.

**13**     Update $\boldsymbol{\mu} \leftarrow \boldsymbol{\mu}' + \alpha(\boldsymbol{\mu} - \boldsymbol{\mu}')$ and $\boldsymbol{\sigma} \leftarrow \boldsymbol{\sigma}' + \alpha(\boldsymbol{\sigma} - \boldsymbol{\sigma}')$.

      // `Send the best evaluated control` $\boldsymbol{u}^* \in \mathbb{R}^n$

**14**     $\boldsymbol{u}^* \leftarrow \boldsymbol{U}_{0,:}$

---

## 4  Experiments

In this section, we investigate our proposed approach with the following research questions/points:

Figure 3: Motivation comparison of MTP methods versus baselines with $B = 256$ on `Navigation` environment. The environment is designed to be challenging to reach the green goal, requiring strong exploration to avoid large local minima in the middle (see task details in Appendix A.3). We plan with $T = 20$ with $\Delta t = 0.05s$. The figures show 5 random traces of white rollouts. (Left) MTP-Akima rollouts reach the green goal very early due to high-entropy tensor sampling, while (Right) OpenAI-ES struggles to generate a rollout exploring the way out of large local minima.

- How does MTP's performance with Akima/B-spline control variants compare to standard sampling-based MPC baselines, and strong-exploratory evolution strategies baselines?

- How does MTP's performance vary with the number of elites and mixing rate $\beta$ on interpolation methods (Linear, B-spline, Akima-spline)?

- How does MTP's performance vary with (i) mixing rate $\beta$ associating with levels of MTP exploration on complex environments, and (ii) MTP-Bspline degree versus planning performance.

We study the cumulative cost $J(\boldsymbol{\tau}, \boldsymbol{U})$ Eq. (2) over the control timestep for each task. For each experiment, we take the minimum cumulative cost in batch rollouts at each timestep, and plot the mean and standard deviation over 5 seeds. Further ablations on sweeping graph parameters $M, N$, `softmax` weighting, and JAX planning performance benchmark are presented in Appendix A.5.

**Practical Settings.** All algorithms and environments are implemented in MuJoCo XLA (Todorov et al., 2012; Kurtz, 2024), to utilize the `jit` and `vmap` JAX operators for efficient vectorized sampling and rollout on multiple model instances. All experiment runs are *sim-to-sim* evaluated (MuJoCo XLA to MuJoCo). In particular, we introduce some modeling errors in MuJoCo XLA. Then, for online domain randomization, we randomize a set of $R$ models $\{f_j(\boldsymbol{x}, \boldsymbol{u})\}_{j=1}^{R}$, then we perform batch sampling and rollout of $R \times B$ trajectories. Finally, the cost evaluation is averaged on the $R$ domain randomization dimension. Note that, for this paper, we deliberately design the task costs to be simple and set sufficiently short planning horizons to benchmark the exploratory capacity of algorithms. In practice, one may design dense guiding costs to achieve the tasks. Further task details are presented in Appendix A.3.

**Baselines.** For explorative baselines, we choose OpenAI-ES (with antithetic sampling) (Salimans et al., 2017), which shows parallelization capability with a high number of workers in high-dimensional model-based RL settings. Additionally, we choose the recently proposed Diffusion Evolution (DE) (Zhang et al., 2024), bridging the evolutionary mechanism with a diffusion process (Ho et al., 2020), which demonstrates superior performances over classical baselines such as CMA-ES (Hansen, 2016). Both are implemented in `evosax` (Lange, 2023). For methods that take into account local information (exploitation methods), we compare MTP with standard MPPI (Williams et al., 2017) and Predictive Sampling (PS) (Howell et al., 2022) to sanity check on task completion in sim-to-sim scenarios.

## 4.1 Motivating Example

Here, we provide an experimental analysis of the baselines' exploration capacity on `Navigation` environment (cf. Fig. 3), where the point-mass agent is controlled by an axis-aligned 2-dim velocity controller. We compare MTP-Bspline and MTP-Akima to evolutionary algorithms and standard MPC baselines, with maximum sampling noise settings (cf. Fig. 3). In particular, given the control limits $[-1, 1]$ on x-y axes, we set the standard deviation $\sigma = 1$ for sampling noise of MPPI and PS, and population generation noise for OpenAI-ES and DE. Fig. 3 also shows the cost convergence and the cost entropy curves over timesteps, in which the entropy is computed as $H = -\sum_{j=1}^{B} P_j \log P_j$, $P_j = \exp(J_j)/(\sum_l \exp(J_l))$, where $\{J_j\}_{j=1}^{B}$ is the batch of cumulative rollout costs. The entropy represents the diversity of rollout evaluation, implying the exploration

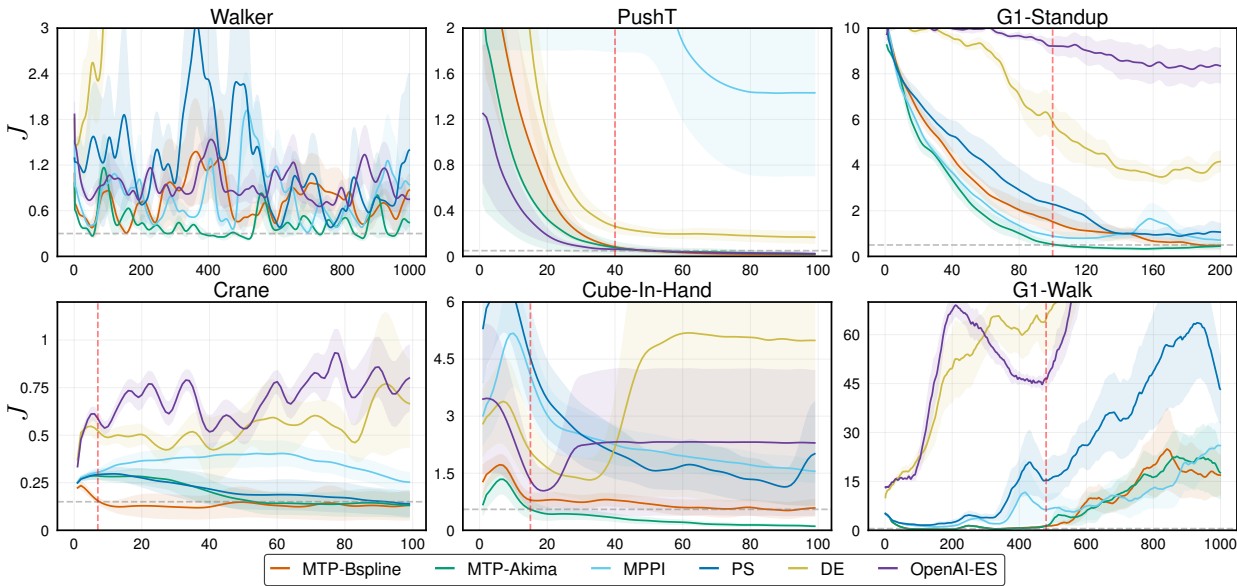

Figure 4: Performance comparison of MTP variants against standard MPC methods (MPPI, PS) and evolutionary algorithms (OpenAI-ES, DE). The (horizontal) gray dashed line depicts the task success, while the (vertical) red line represents the timestep such that the first algorithm statistically succeeds the task, or fails last in `G1-Walk`.

capability of algorithms. We observe that MTP-Akima has the highest entropy curve, inducing the lowest cost convergence over timesteps, while other baselines converge to the middle minima. In this case, MTP-Bspline also struggles due to conservative interpolation in tensor sampling, achieving moderate entropy, yet higher than evolutionary strategies.

## 4.2 Comparison Experiments

We analyze the performance comparison of MTP and the baselines over various robotics tasks representing different dynamics and planning cost settings (cf. Fig. 4). In each task, we tune the baselines and set the same white noise standard deviation for MTP/MPPI/PS to study the performance gain by tensor sampling, while using the default hyperparameters for evolutionary baselines.

**Comparison Environments.** `PushT` (Chi et al., 2023), `Cube-In-Hand` (Andrychowicz et al., 2020) require intricate manipulation environments where robust exploration is critical due to complex contact dynamics and precise multi-step manipulation requirements. `G1-Standup, G1-Walk` represent high-dimensional robotic tasks demanding substantial computational resources and sophisticated control strategies. `Crane`, `Walker` (Towers et al., 2024) present underactuated and nonlinear dynamic challenges. To ensure fair comparison, for all baselines, we fix the same number of rollouts $B = 16$ on `Crane`, and $B = 128$ for all other tasks. All tasks are implemented in `hydrax` (Kurtz, 2024). Further experiment details are in Appendix A.4. The `PushT` task, which involves pushing a T-shaped object precisely to a target location, particularly highlights the advantage of MTP variants. While MPPI and PS frequently encounter mode collapse due to insufficient exploration, resulting in suboptimal or even failed attempts at solving the task, MTP-Bspline and MTP-Akima consistently achieve low-cost convergence. This underscores the significant benefit of strong exploration enabled by tensor sampling. Evolutionary algorithms like OpenAI-ES and DE perform similarly to MTP in `PushT`, inherently show better exploration than MPPI and PS, but still fall short compared to MTP variants in `Cube-In-Hand` due to noisy rollouts. `Cube-In-Hand` requires strong exploration while maintaining intricate control to avoid the cube falling, thus emphasizing the effectiveness of the MTP $\beta$-mixing strategy.

In the `Crane` environment, we apply heavy modeling errors of mass, inertia, and pulley/joint damping. MTP variants excel by maintaining stable control trajectories with smooth transitions, effectively navigating the nonlinear dynamics and underactuation. The B-spline interpolation's conservative nature helps avoid

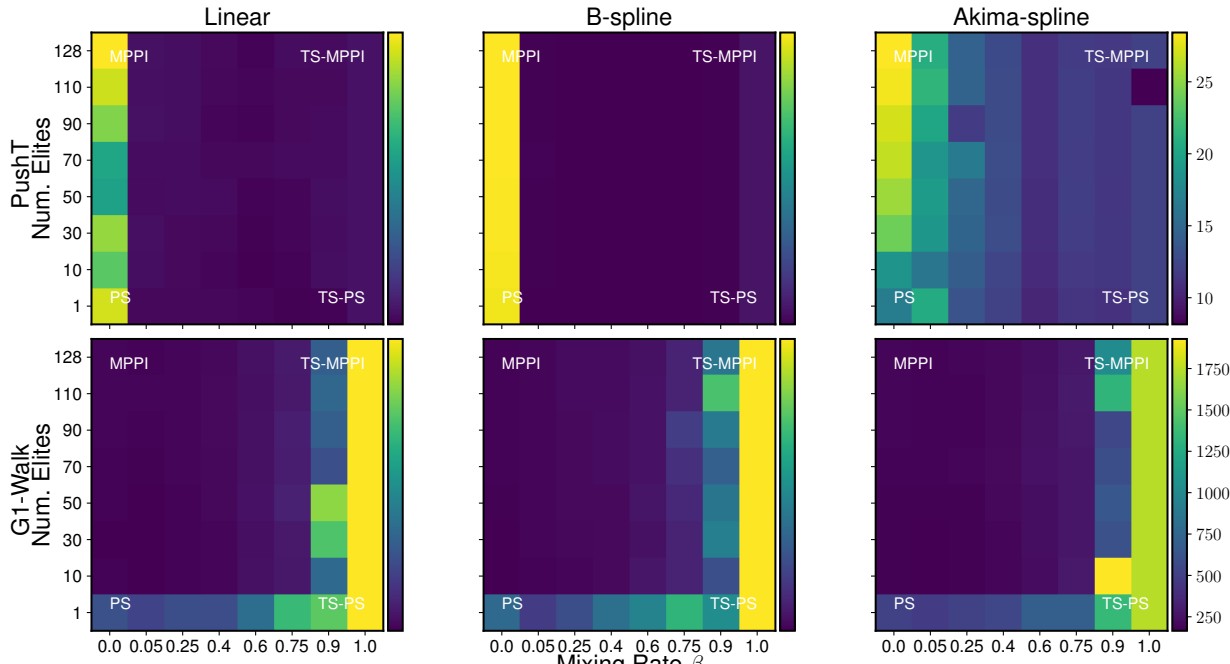

Figure 5: Mixing rate $\beta$ and number of elites $E$ sweep on `PushT`, `G1-Walk` tasks with $B = 128$ to investigate the algorithmic update rule Algorithm 2 Line 9-12. The heatmap indicates accumulated cost over timesteps at termination, and the heat value range is fixed for each task/row.

overshooting and instability prevalent in these tasks, thus outperforming both evolutionary algorithms and standard MPC methods that tend to produce erratic control inputs. The `Walker` task exhibits a rather simple dynamics model and is less sensitive to the sampling distribution due to its relatively simple contact model. We apply no modeling error as a sanity check. Indeed, MTP and classical MPPI/PS perform similarly in simple cases.

In `G1-Standup`, MTP-Akima demonstrates effective humanoid standup due to its aggressive yet smooth trajectory interpolation, enabling efficient exploration and rapid convergence. In contrast, OpenAI-ES and DE struggle with the dimensionality, often yielding higher cumulative costs and failing to adequately sample feasible trajectories, resulting in significant performance gaps. MTP variants have marginally higher performance than standard MPC baselines in `G1-Standup`, but show better control stability for longer `G1-Walk` before falling. These results underline the capability of MTP to balance exploration and exploitation in high-dimensional tasks systematically.

## 4.3 Design Ablation

We conduct an ablation study analyzing the effect of varying two crucial hyperparameters—the number of elites $E$ and the mixing rate $\beta$—on MTP algorithmic performance. Fig. 5 presents heatmaps indicating accumulated cost over time for each task and interpolation method (MTP-Linear, MTP-Bspline, MTP-Akima). Each heatmap illustrates distinct algorithmic realizations at its corners. Specifically, the bottom-left corner represents PS, characterized by a single elite and purely white noise sampling. Conversely, the bottom-right corner corresponds to Predictive Tensor Sampling (TS-PS), maintaining a single elite but employing full tensor sampling. Due to the `softmax` update (Algorithm 2 Line 11-12), the top-left corner realizes MPPI (with adaptive sampling covariance), leveraging all candidate samples with local noise sampling, while the top-right corner reflects Tensor Sampling-MPPI (TS-MPPI), utilizing all candidates and full tensor sampling for maximum exploration.

We observe the consistent pattern that MTP performance degrades at the extremes of $\beta$. In `PushT`, moderate values of $\beta$ lead to significantly lower costs, while the absence of tensor sampling ($\beta = 0$) yields poor performance due to inadequate exploration. This observation reinforces the effectiveness of the $\beta$-mixing

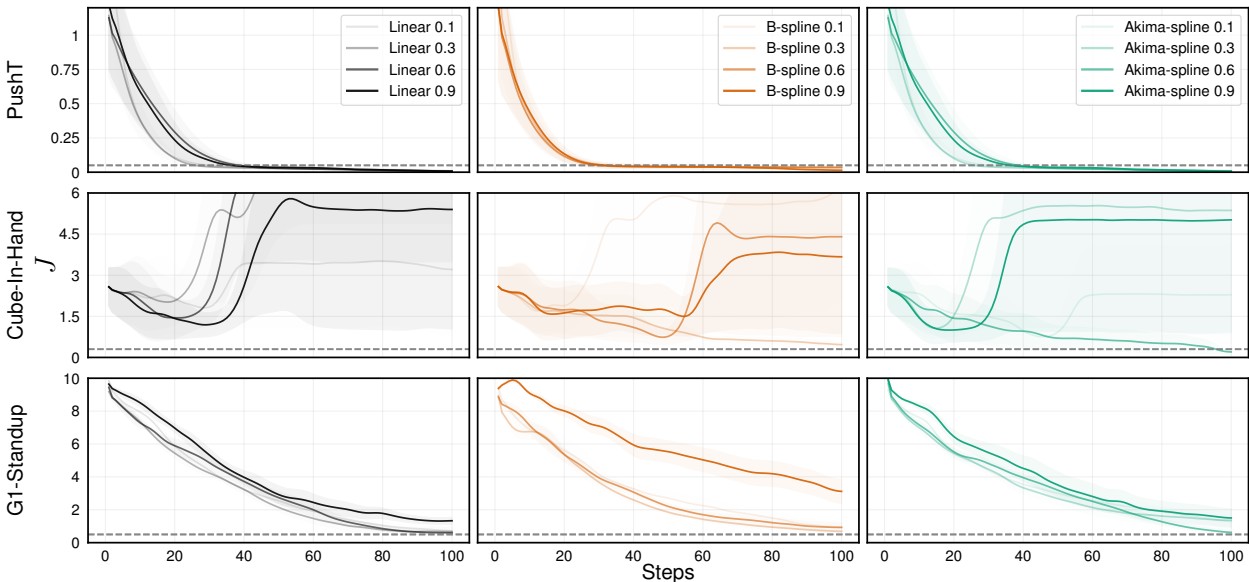

Figure 6: Mixing scalar $\beta$ sweep on `PushT, Cube-In-Hand, G1-Standup` environments with $B = 128$ to investigate the sensitivity of MTP on explorative level. The dashed line represents the successful bar. In `Cube-In-Hand`, some of the cost curves increase due to the cube falling out of the LEAP hand.

strategy for balancing global and local sampling contributions. Interestingly, the number of elites $E$ has a limited impact in `PushT`, likely due to the task's insensitivity to control stability. However, in `G1-Walk`, the choice of $E$ is crucial. Using a single elite ($E = 1$), corresponding to the PS control scheme, leads to unstable and jerky behavior, which aligns with the poor performance observed in Fig. 4. At the other extreme, with $\beta = 1$ (full tensor sampling), performance also degrades. This is attributed to the fixed rollout budget $B$; as $M$ increases, global samples become too sparse to effectively capture the fine-grained control required for stable gait tracking. In this case, exploitation with local samples is essential to maintain intricate motion tracking control.

### 4.4 Sensitivity Ablation

Here, we perform sensitivity analyses for various critical algorithmic hyperparameters. Fig. 6 evaluates sensitivity to the mixing rate $\beta$ for different tasks. For the `PushT` environment, results show minimal sensitivity across mixing rates, as the task inherently lacks significant failure modes, ensuring consistent success regardless of the exploration-exploitation balance. In contrast, the `Cube-In-Hand` task demonstrates high sensitivity, with larger mixing rates causing instability due to the cube falling out of grasp frequently. Optimal performance is thus achieved with lower $\beta$ values, suggesting careful management of exploration intensity. Furthermore, for the high-dimensional `G1-Standup` task, a smaller mixing rate helps stabilize the control, enabling the robot to achieve a more consistent and stable stand-up performance.

## 5 Related Works

We review related efforts across two major directions: the vectorization of sampling-based MPC and sampling-based motion planning. While these approaches originate from different planning paradigms, dynamics-aware MPC versus collision-free geometric planning, they share a common structure: interacting with an agent-environment model (e.g., dynamics model or collision checker) that can be vectorized for efficient, batched computation. Both domains benefit from high-throughput sampling, making them increasingly amenable to modern GPUs/TPUs.

**Sampling-based MPC Vectorization.** Sampling-based MPC (Mayne, 2014) has been successfully applied to high-dimensional, contact-rich control problems. Methods such as Predictive Sampling (PS) (Howell et al., 2022), Model Predictive Path Integral (MPPI) (Williams et al., 2017; Watson & Peters, 2023), and

CEM-based MPC (Pinneri et al., 2021) rely on parallel sampling of control trajectories and subsequent rollouts using a system dynamics model. These methods naturally benefit from vectorized simulation backends, and recent works have extended them toward more structured and efficient exploration. For instance, inspired by the diffusion process, DIAL-MPC (Xue et al., 2024) enhances exploration coverage and local refinement simultaneously, achieving high-precision quadruped locomotion and outperforming reinforcement learning policies (Schulman et al., 2017) in climbing tasks. STORM (Bhardwaj et al., 2022) demonstrates GPU-accelerated joint-space MPC for robotic manipulators, achieving real-time performance while handling task-space and joint-space constraints. Other recent efforts integrate GPU-parallelizable simulators, such as IsaacGym (Makoviychuk et al., 2021), into the MPC loop (Pezzato et al., 2025) for online domain randomization, removing the need for explicit modeling and enabling real-time contact-rich control. In another line, CoVO-MPC (Yi et al., 2024) enhances convergence speed by optimizing the covariance matrix during sampling, leading to performance gains in both simulated and real-world quadrotor tasks. These advances demonstrate that structured, parallel control sampling can be effectively deployed in high-stakes robotics applications using vectorized dynamic models.

**Motion Planning Vectorization.** Recent advances in sampling-based motion planning have demonstrated that classical methods, such as RRT (Kuffner & LaValle, 2000), can be significantly accelerated through parallel computation, while preserving theoretical guarantees like probabilistic completeness, which represents another form of maximum exploration. Early work focused on accelerating specific subroutines like collision checking (Bialkowski et al., 2011; Pan & Manocha, 2012), but more recent efforts have restructured planners for full GPU-native execution. Examples include GMT* (Lawson et al., 2020), VAMP (Thomason et al., 2024), pRRTC (Huang et al., 2025), and Kino-PAX (Perrault et al., 2025), which achieve millisecond-scale planning in high-dimensional configuration spaces by parallelizing sampling, forward kinematics, and tree expansions. GTMP (Le et al., 2025) pushes this even further by implementing the sampling, graph-building, and search pipeline as tensor operations over batch-planning instances, showcasing the feasibility of real-time planning across multiple environments.

Complementing sampling-based planning, trajectory optimization methods such as batch CHOMP (Zucker et al., 2013), Stochastic-GPMP (Urain et al., 2022), cuRobo (Sundaralingam et al., 2023), and MPOT (Le et al., 2023) have embraced vectorization to solve hundreds of trajectory refinement problems in parallel. Many of these systems are further enhanced by high-entropy initialization with learned priors (Carvalho et al., 2023; Huang et al., 2024; Nguyen et al., 2025), allowing them to overcome challenging nonconvexities in cluttered environments. These developments collectively demonstrate that both motion planning and MPC can be reformulated as batched, tensor-based pipelines suitable for modern accelerators.

Our work draws on these insights to propose a unified sampling-based control framework that operates entirely through tensorized computation, blending global exploration and local refinement in a single batched planning loop.

## 6 Discussions and Conclusions

In this work, we introduced *Model Tensor Planning* (MTP), a robust sampling-based MPC approach designed to achieve global exploration via maximum entropy sampling. Theoretically, we demonstrated that in the limits of infinite layers $M$ and samples per layer $N$, our tensor sampling method attains maximum entropy, thereby efficiently approximating the full trajectory space. Furthermore, MTP is intentionally designed to be practically feasible, enabling straightforward implementation for sampling high-entropy control trajectories (see Appendix A.5).

While evolutionary strategies algorithms offer improved exploration capabilities (Salimans et al., 2017; Zhang et al., 2024), compared to traditional MPC methods, our experiments highlight their limitations. The inherently noisy mutation processes often fail to achieve consistent high-entropy exploration, limiting their effectiveness in robotics tasks. In contrast, MTP's tensor sampling consistently explores smooth control possibilities and achieves robust performance.

Spline-based interpolations are central to the practical implementation of MTP, notably B-spline and Akima-spline. These interpolation methods effectively address discontinuities in simple linear interpolation, ensuring

the generation of smooth, continuous, and dynamically feasible control trajectories (Alvarez-Padilla et al., 2024). The experiments underscore the splines' critical role in enhancing trajectory quality, optimizing performance across diverse, complex tasks. We proposed a simple $\beta$-mixing strategy for exploration while retaining intricate controls, effectively balancing exploration and exploitation within sampling-based MPC. This flexible strategy allows easy algorithm tuning to various tasks, significantly improving performance stability and robustness across environments with different exploration needs.

From the vectorization standpoint, the matrix-based definition of MTP is specifically structured to leverage Just-in-time compilation `jit` and vectorized mapping `vmap` provided by JAX (Bradbury et al., 2018) and MuJoCo XLA (Todorov et al., 2012). This design choice dramatically accelerates computations, enabling real-time implementation and seamless integration with online domain randomization with `vmap`, crucial for robust control. Overall, MTP offers an efficient, scalable solution for various robotic tasks that demand high exploration capacity and precise control optimization.

## Limitations

While MTP demonstrates strong performance across diverse control tasks, it inherits several limitations typical of sampling-based methods. First, its computational cost scales with the number of rollouts, making it challenging to deploy on hardware with limited parallel computing. Second, while our tensor-based sampler improves exploration coverage, it does not leverage task-specific priors or learning-based proposal distributions, which could further improve sample efficiency. Finally, MTP relies on a fixed dynamics model, limiting its robustness in partially observed or stochastic environments.

## Broader Impact Statement

This work contributes to the development of efficient sampling-based control by introducing a scalable, high-entropy sampling mechanism for model predictive control (MPC). *Model Tensor Planning* (MTP) opens a promising direction in the design of exploratory algorithms that go beyond local refinements, allowing for global reasoning over control spaces. By enabling maximum entropy exploration via structured tensor operations, MTP provides a framework that may benefit a wide range of decision-making systems requiring robust performance in underexplored, high-dimensional environments, such as dexterous manipulation, legged locomotion, or autonomous vehicles operating under partial observability and uncertainty.

From a real-world deployment perspective, MTP maintains controls within physical limits, but its high-rate, tensorized control sequences may induce rapid variations that practical motor systems must robustly execute. While this fast-changing control is beneficial for agility and responsiveness, it necessitates attention to actuator dynamics and hardware safety. Therefore, safety-aware control filtering or actuator-aware smoothness constraints may be incorporated as extensions for deployment.

Furthermore, like other MPC approaches, MTP assumes access to reliable state estimation for initializing planning rollouts in the control loop. In practical deployment, this typically requires a real-time state estimator and a simulation back-end that serves as a digital twin. For example, MuJoCo XLA can simulate hundreds of dynamics instances in parallel, making it suitable for real-time predictive control. However, realizing this in hardware introduces engineering challenges, such as ensuring low-latency communication between the physical robot and the simulator. We see this digital twin architecture as a promising frontier where algorithmic advances like MTP can be tightly integrated with system-level design for robust, real-time, and scalable autonomous control.

### Acknowledgments

This work was funded by the German Federal Ministry of Education and Research Software Campus project ROBOSTRUCT (01S23067).

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

# A   Appendix

## A.1   Proof of Theorem 1

Let $u \in \mathbb{F}$ be any path that is uniformly continuous and has bounded variation $TV(u) < \infty$. We begin by constructing a piecewise linear path approximating $u$, $g_M : [0, 1] \to \mathbb{U}$ by dividing the interval $[0, 1]$ into $M - 1$ subintervals, i.e., $[t_1, t_2], \ldots, [t_{M-1}, t_M]$ with $0 = t_1 < t_2 < \ldots < t_M = 1$. On each subinterval $[t_i, t_{i+1}]$, we define the corresponding segment of $g$ to approximate $u$

$$\boldsymbol{g}(t) = \boldsymbol{u}(t_i) + \frac{\boldsymbol{u}(t_{i+1}) - \boldsymbol{u}(t_i)}{t_{i+1} - t_i}(t - t_i), \ t \in [t_i, t_{i+1}]. \tag{11}$$

Then, we define a control path in $\mathcal{G}(M, N)$.

**Definition 4** (Path In $\mathcal{G}(M, N)$). *A path $u : [0, 1] \to \mathbb{U}$ is in $\mathcal{G}(M, N)$ (i.e., $u \in \mathcal{G}(M, N)$) if and only if $u$ is piecewise linear having $M - 1$ segments and $\forall 1 \le i \le M, \boldsymbol{u}(t_i) \in L_i$.*

**Lemma 1** (Piecewise Linear Path Approximation). *Let $g_1, g_2$ be a piecewise linear function having the same number of partition points $\{\boldsymbol{g}_1(t_i)\}_{i=1}^M, \{\boldsymbol{g}_2(t_i)\}_{i=1}^M$ with $0 = t_1 < t_2 < \ldots < t_M = 1$, $\|g_1 - g_2\|_\infty < \epsilon$, if and only if $\|\boldsymbol{g}_1(t_i) - \boldsymbol{g}_2(t_i)\| < \epsilon$, $1 \le i \le M$.*

*Proof.* **Sufficiency.** Given $\|\boldsymbol{g}_1(t_i) - \boldsymbol{g}_2(t_i)\| < \epsilon$, $\forall 1 \le i \le M$, since $g_1, g_2$ are piecewise linear functions, the linear interpolation between partition points $t_i, t_{i+1}$ ensures that the difference between $g_1, g_2$ is maximized at the partition points. Consider $g_1, g_2$ on a segment $[t_i, t_{i+1}]$

$$\|\boldsymbol{g}_1(t) - \boldsymbol{g}_2(t)\| \le \max\{\|\boldsymbol{g}_1(t_i) - \boldsymbol{g}_2(t_i)\|, \|\boldsymbol{g}_1(t_{i+1}) - \boldsymbol{g}_2(t_{i+1})\|\} < \epsilon \tag{12}$$

Hence, $\|g_1 - g_2\|_\infty = \max_{t \in [0,1]} \|\boldsymbol{g}_1(t) - \boldsymbol{g}_2(t)\| < \epsilon$.

**Necessity.** Given $\|g_1 - g_2\|_\infty < \epsilon$, then $\|\boldsymbol{g}_1(t_i) - \boldsymbol{g}_2(t_i)\| < \epsilon$, $1 \le i \le M$.   $\square$

Now, we investigate that any piecewise linear path $g$ with $M - 1$ equal subintervals, approximating $u \in \mathbb{F}$, uniformly converges to $u$ as $M \to \infty$.

**Lemma 2** (Convergence Of Linear Path Approximation). *Let $g_M$ be any piecewise linear path approximating $u \in \mathbb{F}$ having $M$ equal subintervals of width $h = 1/M$. Then,*

$$\lim_{M \to \infty} \|u - g_M\|_\infty = 0.$$

*Proof.* Since $u$ is uniformly continuous on $[0, 1]$, for any $\epsilon > 0$, there exists $\delta > 0$ such that for all $t, s \in [0, 1]$, if $|t - s| < \delta$, then

$$\|\boldsymbol{u}(t) - \boldsymbol{u}(s)\| < \frac{\epsilon}{2}. \tag{13}$$

Also, the variation of $u$ within each subinterval approaches zero as $M \to \infty$ due to the uniformly continuous property. Hence, for sufficiently large $M$, each subinterval length $h = \frac{1}{M} < \delta$, and thus:

$$\sup_{t \in [t_i, t_{i+1}]} \|\boldsymbol{u}(t) - \boldsymbol{g}_m(t)\| = \sup_{t \in [t_i, t_{i+1}]} \left\| \boldsymbol{u}(t) - \boldsymbol{u}(t_i) + \frac{\boldsymbol{u}(t_{i+1}) - \boldsymbol{u}(t_i)}{h}(t - t_i) \right\|$$

$$\le \sup_{t \in [t_i, t_{i+1}]} \|\boldsymbol{u}(t) - \boldsymbol{u}(t_i)\| + \sup_{t \in [t_i, t_{i+1}]} \left\| \frac{\boldsymbol{u}(t_{i+1}) - \boldsymbol{u}(t_i)}{h}(t - t_i) \right\| < \frac{\epsilon}{2} + \frac{\epsilon}{2} = \epsilon. \tag{14}$$

Taking the supremum over all $t \in [0, 1]$ (i.e., over $M$ equal subintervals), we obtain:

$$\|u - g_M\|_\infty < \epsilon. \tag{15}$$

Since $\epsilon > 0$ is arbitrary and $h \to 0$ as $M \to \infty$, it follows that:

$$\lim_{M \to \infty} \|u - g_M\|_\infty = 0. \tag{16}$$

$\square$

We now prove that the random multipartite graph discretization is asymptotically dense in $\mathbb{F}$. Specifically, as the number of layers $M$ and the number of samples per layer $N$ approach infinity, the graph will contain a path that uniformly approximates any continuous path in $\mathbb{F}$.

**Theorem 1** (Asymptotic Path Coverage). *Let $u \in \mathbb{F}$ be any control path and $\mathcal{G}(M, N)$ be a random multipartite graph with $M$ layers and $N$ uniform samples per layer (cf. Definition 1). Assuming a time sequence (i.e., knots) $0 = t_1 < t_2 < \ldots < t_M = 1$ with equal intervals, associating with layers $L_1, \ldots, L_M \in \mathcal{G}(M, N)$ respectively, then*

$$\lim_{M,N \to \infty} \min_{g \in \mathcal{G}(M,N)} \|u - g\|_\infty = 0.$$

*Proof.* First, Lemma 2 implies that there exists a sequence of linear piecewise $g_M$, having $M - 1$ equal intervals approximating $u$, converging to $u$ as $M \to \infty$.

Let $\hat{g}_M \in \mathcal{G}(M, N)$ be a control path in $\mathcal{G}$ (cf. Definition 4). Since the time sequence $0 = t_1 < t_2 < \ldots < t_M = 1$ corresponding to layers $L_1, \ldots, L_M \in \mathcal{G}(M, N)$ has equal intervals, we can consider $\hat{g}_M$ having $M - 1$ segments approximating $g_M$ without loss of generality.

Since $\mathbb{U}$ is open, for each $i = 1, \ldots, M$, there exists a ball $B_\epsilon(\boldsymbol{u}(t_i)) \subset \mathbb{U}$, $\epsilon > 0$. By definition $\hat{g}_M, g_M$ has the same number of segments, the event $\|\hat{g}_M - g_M\|_\infty < \epsilon$ is the event that, for each layer $1 \le i \le M$, there is at least one point $\boldsymbol{g}_M(t_i)$ is sampled inside the ball $B_\epsilon(\hat{\boldsymbol{g}}_M(t_i))$. The probability that none of the $N$ samples in layer $L_i$ fall inside $B_\epsilon(\boldsymbol{g}_M(t_i))$ is

$$\left(1 - \frac{\mu(B_\epsilon(\boldsymbol{u}(t_i)))}{\mu(\mathbb{U})}\right)^N \le e^{-cN} \tag{17}$$

for some $c > 0$. From Lemma 1, the probability that every layer contains at least one such sample, such that $\|g_M - \hat{g}_M\|_\infty < \epsilon$, is at least $1 - Me^{-cN}$, which converges to 1 as $N \to \infty$.

From Lemma 2, for sufficiently large $M$, we have $\|u - g_M\|_\infty < \epsilon$. Now, due to $\mathbb{U}$ is compact, we can apply the triangle inequality as $N \to \infty$

$$\|u - \hat{g}_M\|_\infty \le \|u - g_M\|_\infty + \|g_M - \hat{g}_M\|_\infty < \epsilon + \epsilon = 2\epsilon. \tag{18}$$

Since $\epsilon$ was arbitrary, we conclude that

$$\lim_{M,N \to \infty} \min_{\hat{g}_M \in G(M,N)} \|u - \hat{g}_M\|_\infty = 0. \tag{19}$$

$\square$

## A.2 Exploration Versus Exploitation Discussion

We investigate the tensor sampling Definition 1 (with linear interpolation) versus MPPI sampling with horizon $T$, corresponding to global exploration versus local exploitation behaviors from the current system state. In particular, we remark on the entropy of path distributions in both methods in the discretized control setting with equal time intervals. Further investigation on the continuous control setting is left for future work.

Let a discrete control path be $\boldsymbol{\tau} = [\boldsymbol{u}_1, \ldots, \boldsymbol{u}_T] \in \mathbb{R}^{T \times n}, \boldsymbol{u} \in \mathbb{U}$. Let $P(\boldsymbol{\tau})$ be the probability of sampling control path $\boldsymbol{\tau}$ under a given planning method. The entropy is then defined as

$$H(P) = -\sum_{\boldsymbol{\tau} \in \mathbb{F}_T} P(\boldsymbol{\tau}) \log P(\boldsymbol{\tau}), \tag{20}$$

where $\mathbb{F}_T \subset \mathbb{F}$ is the set of all possible discrete control paths $\boldsymbol{\tau}$ of length $T$.

Consider $\mathcal{G}(M, N)$, each node in layer $L_i$ is sampled independently from a uniform distribution over $\mathbb{U}$, and path candidates are equivalent to sequences of node indices $\boldsymbol{\tau} \sim \tau = (i_1, i_2, \ldots, i_M) \in \{1, \ldots, N\}^M$ (cf. Algorithm 1). Let $\mathcal{S}$ denote the set of all index sequences representing valid paths through the graph.

The uniform distribution over $\mathcal{S}$ is given by $P_{\mathcal{G}}(\boldsymbol{\tau}) = 1/|\mathcal{S}|$, where $|\mathcal{S}| = N^M$. Hence, the entropy of tensor sampling is

$$H(P_{\mathcal{G}}) = -\sum_{\tau \in \mathcal{S}} (1/N^M) \log(1/N^M) = \log(N^M) = M \log N. \qquad (21)$$

Indeed, as $M, N \to \infty$, the entropy $H(P_{\mathcal{G}}) \to \infty$, and the distribution over sampled paths in $\mathcal{G}$ becomes maximum entropy over $\mathbb{F}_T$ among all discrete path distributions. Theorem 1 implies that $\mathbb{F}_T \to \mathbb{F}$ as $M, N \to \infty$, and thus tensor sampling distribution becomes maximum entropy over $\mathbb{F}$.

Now, typical MPPI implementation generates control paths by perturbing a nominal trajectory $\bar{\boldsymbol{\tau}}$ with Gaussian noise (Vlahov et al., 2024) $\boldsymbol{u}_t = \bar{\boldsymbol{u}}_t + \boldsymbol{\epsilon}_t$, $\boldsymbol{\epsilon}_t \sim \mathcal{N}(\mathbf{0}, \boldsymbol{\Sigma})$, and propagating the dynamics to generate a state trajectory. The path distribution $P_{\text{MPPI}}$ concentrates around $\bar{\boldsymbol{\tau}}$ and is generally non-uniform. The entropy is constant and computed in closed form

$$H(P_{\text{MPPI}}) = \frac{Tn}{2}(1 + \log(2\pi)) + T \log \det(\boldsymbol{\Sigma}), \qquad (22)$$

due to independent Gaussian noise over timestep (i.e., white noise kernel (Watson & Peters, 2023)).

In general, tensor sampling serves as a configurable high-entropy sampling mechanism over control trajectory space, offering maximum exploration, while MPPI targets local improvement around a nominal trajectory, thereby performing exploitation. This distinction motivates the hybrid method, where we mix explorative (smooth) controls with local controls sampled from a typical white noise kernel.

### A.3 Task Details

Here, we provide task details on the task, cost definitions, and their domain randomization. There exist motion capture sensors in MuJoCo to implement the tasks. For this paper, we deliberately design the task costs to be simple and set sufficiently short planning horizons to benchmark the exploratory capacity of algorithms. In practice, one may design dense guiding costs to make tasks easier.

**Navigation.** A planar point mass moves in a bounded 2D space via velocity commands to reach a target while avoiding collisions. The state cost is defined as

$$c(\boldsymbol{x}_t, \boldsymbol{u}_t) = \alpha_1 \exp(-\lambda d_{\text{wall}}(\boldsymbol{x}_t) + \alpha_2 \|\boldsymbol{x}_t - \boldsymbol{x}_g\|^2 + \alpha_3 \|\boldsymbol{u}_t\|^2 ,$$

where $d_{\text{wall}}(\boldsymbol{x}_t)$ is the distance to the closest wall, $\boldsymbol{x}_g$ is the goal position, and $\boldsymbol{u}_t$ is the velocity control. Success is defined when the agent's distance to the target is sufficiently small. This task is extremely difficult for sampling-based MPC due to large local minima near the starting point.

**Crane.** The agent controls a luffing crane via torque inputs to move a suspended payload to a target while minimizing oscillations. The cost function penalizes payload deviation and swing:

$$c(\boldsymbol{x}_t, \boldsymbol{u}_t) = \alpha_1 \|\boldsymbol{x}_t - \boldsymbol{x}_g\|^2 + \alpha_2 \|\dot{\boldsymbol{x}}_t\|^2 ,$$

where $\boldsymbol{x}_t$ is the payload tip position, $\boldsymbol{x}_g$ is the target point, and $\dot{\boldsymbol{x}}_t$ is the tip velocity. Success is achieved when the payload tip is within a small radius of the target location. This task is difficult due to heavy modeling error and underactuation, which is common in real crane applications.

**Cube-In-Hand.** Using velocity control of a dexterous LEAP hand, the agent must rotate a cube to match a randomly sampled target orientation. The cost combines position and orientation error:

$$c(\boldsymbol{x}_t, \boldsymbol{u}_t) = \alpha_1 d_{\text{SE}(3)}(\boldsymbol{x}_t, \boldsymbol{x}_g)^2 + \alpha_2 \|\dot{\boldsymbol{x}}_t\|^2 ,$$

where $\boldsymbol{x}_t$ and $\boldsymbol{x}_g$ are the current cube and target poses, $d_{\text{SE}(3)}$ is the SE(3) distance metric between poses, and $\boldsymbol{u}_t = \dot{\boldsymbol{x}}_t$. Success is defined when the combined position and orientation errors fall below a threshold. This task is difficult due to the high-dimensional, contact-rich, and failure mode of the falling cube.

**G1-Walk.** A Unitree G1 humanoid robot tracks a motion-captured walking trajectory using position control. The cost is defined as the deviation from reference joint positions:

$$c(\boldsymbol{x}_t, \boldsymbol{u}_t) = \alpha_1 \|\boldsymbol{x}_t - \boldsymbol{x}_{\text{ref}}(t + k)\|^2 ,$$

where $\boldsymbol{x}_t$ and $\boldsymbol{x}_{\mathrm{ref}}(t+k)$ are the current joint and reference joint positions, given the current control iteration $k$. Success is not binary but is measured by minimizing deviation from the reference joint configurations. Note that this cost is not designed for stable locomotion. The main challenge is maintaining motion tracking locomotion over long horizons with complex joint couplings.

**G1-Standup.** The humanoid must rise from a lying pose to an upright standing posture. The cost penalizes deviation from upright pose and instability:

$$c(\boldsymbol{x}_t, \boldsymbol{u}_t) = \alpha_1 (h_t - h^*)^2 + \alpha_2 d_{\mathrm{SO}(3)}(\boldsymbol{R}_{\mathrm{torso}}, \boldsymbol{R}_g)^2 + \alpha_3 \left\| \boldsymbol{q}_t - \boldsymbol{q}_{\mathrm{nominal}} \right\|^2,$$

where $h_t, h^*$ is the torso height and the standing height threshold, $\boldsymbol{R}_t^{\mathrm{torso}}, \boldsymbol{R}_g$ are the orientation of current and target torso. $h_t, \boldsymbol{R}_t^{\mathrm{torso}}, \boldsymbol{q}_t$ are elements of $\boldsymbol{x}_t$. Success is defined when the height of the torso exceeds a target threshold. The task is difficult due to large initial instability and the need to achieve balance in high-dimensional dynamics.

**PushT.** A position-controlled end effector to push a T-shaped block to a goal pose. The cost measures block pose error

$$c(\boldsymbol{x}_t, \boldsymbol{u}_t) = \alpha_1 d_{\mathrm{SE}(3)}(\boldsymbol{x}_t, \boldsymbol{x}_g)^2,$$

where $\boldsymbol{x}_t$ and $\boldsymbol{x}_g$ are the current T-block and target poses. Success is achieved when the block's position and orientation errors are minimized. The task is challenging because contact dynamics is complex, requiring precise interaction strategies.

**Walker.** A planar biped must walk forward at a desired velocity while maintaining an upright torso. The cost function penalizes deviation from target velocity and orientation:

$$c(\boldsymbol{x}_t, \boldsymbol{u}_t) = \alpha_1 (h_t - h^*)^2 + \alpha_2 (\theta_t - \theta^*) + \alpha_3 (v_t - v^*)^2 + \alpha_4 \left\| \boldsymbol{u}_t \right\|^2,$$

where $h_t, h^*$ is the torso height and the standing height threshold, $v_t, v^*$ is the forward and target velocity, $\theta_t, \theta^*$ is the torso and target pitch angle. $h_t, \theta_t, v_t$ are elements of $\boldsymbol{x}_t$. Success is measured by stable forward motion and velocity tracking. The difficulty lies in generating stable gaits without explicit foot placement planning.

Table 1 summarizes the environments used in our experiments, including their state and action space dimensions, control modalities, and whether domain randomization or task randomization was applied. We set the control horizon and sim step per plan such that they resemble realistic control settings.

Table 1: Summary of environment properties.

| Task | State Dim | Action Dim | Control Type | Domain Randomization |
|---|---|---|---|---|
| Navigation | 4 | 2 | Velocity | Joint obs. noise, actuation gain, init position |
| Crane | 24 | 3 | Torque | Payload mass, inertia, joint damping, actuation gain |
| Cube-In-Hand | 39 | 16 | Velocity | Joint obs. noise, geom friction |
| G1-Walk | 142 | 29 | Position | Joint obs. noise, geom friction |
| G1-Standup | 71 | 29 | Position | Joint obs. noise, geom friction |
| PushT | 14 | 2 | Position | Geom friction, init pose |
| Walker | 18 | 6 | Torque | None (fixed init) |

## A.4 Experiment Details & Discussions

**Comparison Experiment.** We summarize the simulation settings for comparison experiments (cf. Section 4.2) in Table 2, and the hyperparameters used for MTP variants in Table 3. Tables 4 summarize the hyperparameters used for MPPI and PS (temperature is not relevant for PS). The same noise $\sigma$ is used for MTP local samples. For evolutionary strategies (DE and OpenAI-ES), we use default hyperparameters in `evosax` (Lange, 2023).

**Design Ablation.** We conducted the sweep on $\beta \in [0, 1]$ and the number of elites to experimentally study the algorithmic design. The MTP hyperparameters in Section 4.3 are the same as in Table 3. Each setting was evaluated with 4 random seeds.

Table 2: Simulation Settings for Experiments

| Task | Horizon $\Delta t$ [s] | Horizon | Sim Step/Plan | Sim Hz | Num. Randomizations |
|------|------|------|------|------|------|
| Navigation | 0.05 | 20 | 2 | 100 | 8 |
| Crane | 0.4 | 2 | 16 | 500 | 32 |
| Cube-In-Hand | 0.04 | 3 | 2 | 100 | 8 |
| G1-Walk | 0.1 | 4 | 1 | 100 | 4 |
| G1-Standup | 0.2 | 3 | 1 | 100 | 4 |
| PushT | 0.1 | 5 | 10 | 1000 | 4 |
| Walker | 0.15 | 4 | 15 | 200 | 1 |

Table 3: MTP Hyperparameters

| Task | M | N | $\sigma_{\min}$ | Elites | $\beta$ | $\alpha$ |
|------|------|------|------|------|------|------|
| Navigation | 5 | 30 | - | - | 1.0 | - |
| Crane | 2 | 30 | 0.05 | 8 | 0.5 | 0.0 |
| Cube-In-Hand | 2 | 50 | 0.15 | 5 | 0.5 | 0.1 |
| G1-Standup | 2 | 100 | 0.2 | 100 | 0.05 | 0.0 |
| G1-Walk | 2 | 100 | 0.1 | 100 | 0.02 | 0.0 |
| PushT | 3 | 50 | 0.1 | 20 | 0.5 | 0.0 |
| Walker | 2 | 50 | 0.3 | 20 | 0.5 | 0.5 |

Table 4: PS/MPPI Hyperparameters

| Task | Noise Std. $\sigma$ | Temperature |
|------|------|------|
| Navigation | 1.0 | 0.1 |
| Crane | 0.05 | 0.1 |
| Cube-In-Hand | 0.15 | 0.1 |
| G1-Standup | 0.2 | 0.1 |
| G1-Walk | 0.1 | 0.01 |
| PushT | 0.3 | 0.1 |
| Walker | 0.3 | 0.1 |

**Sensitivity Ablation.** We conducted $\beta$-mixing rate ablation study (cf. Section 4.4) by varying the $\beta$ across different MTP variants. The MTP hyperparameters in Section 4.4 are the same as in Table 3. Each configuration was evaluated with 4 random seeds.

**Experimental Discussions.** In tasks with well-shaped or dense reward structures and moderately non-linear dynamics, exploration becomes less critical and nominal sampling methods (e.g., MPPI, PS) often suffice—explaining MPPI's strong performance in G1-Standup, which benefits from fully actuated dynamics and informative rewards. However, in more challenging scenarios involving sparse rewards or highly non-linear dynamics, such as PushT and under domain shifts in Crane, these locally guided strategies tend to struggle with inadequate exploration, often converging to suboptimal solutions or showing high performance variance. A representative case is the Navigation task (Figure 3), where the agent must discover velocity sequences to bypass obstacles and reach the goal—a setting in which local Gaussian sampling clearly fails by getting trapped in local minima. To address these challenges, MTP introduces a structured high-entropy sampling mechanism along with a simple yet effective $\beta$-mixing strategy that balances global exploration and local exploitation. With careful tuning of $\beta$, $M$, and $N$, MTP demonstrates robust and consistent performance across diverse tasks.

**Mixing Rate Tuning.** $\beta$ determines the ratio between exploratory (tensor sampling) and exploitative (nominal sampling) samples, which is delicate to tune. As shown in our ablations in Fig. 6 and Fig. 5, high $\beta$ values (e.g., 0.5) introduce strong exploration, which may benefit tasks with sparse rewards and not requiring delicate fixed-point stability (e.g., PushT, Cube-In-Hand, Navigation). Conversely, in tasks requiring high stability or precise actuation, such as G1-Standup or G1-Walk, lower $\beta$ values (e.g., 0.05-0.1) tend to yield better performance by favoring consistent behavior while still injecting enough exploration to escape suboptimal solutions.

## A.5 Additional Ablation & Performance Benchmarks

In this section, we conduct more MTP ablations to understand how hyperparameters affect MTP performance, and also briefly benchmark the baseline JAX implementations to confirm the real-time performance.

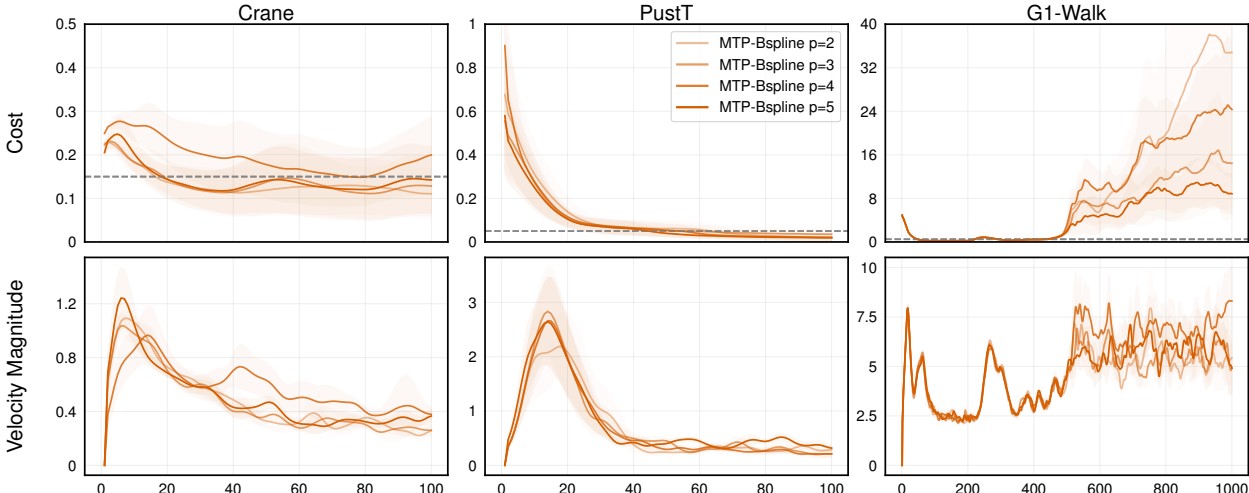

Figure 7: MTP-Bspline degree ablation. In `G1-Walk`, the Unitree G1 controlled Bspline degrees all roughly fall at 500 time steps.

**B-spline Degree Ablation.** We investigate the sensitivity of MTP performance over B-spline interpolation degrees. The MTP hyperparameters are the same as in Table 3. Each setting was evaluated using 4 random seeds.

In Fig. 7, we investigate the sensitivity of MTP performance over B-spline interpolation degrees. Results consistently show minimal performance differences across degrees ($p = 2$ to $p = 5$) in terms of both cost and velocity magnitude curves across different tasks. Given this insensitivity, we select the lower computational complexity B-spline $p = 2$ as our default choice for the MTP-Bspline method.

**Sweep $M, N$ Ablation.** We conducted an additional ablation to study the effect of varying $M$ and $N$ values of MTP variants on the `Navigation` task. We set $\beta = 1$ to use full tensor sampling. Each configuration was evaluated with 4 random seeds. According to Fig. 8, there exists a sweet point in selecting the number of layers

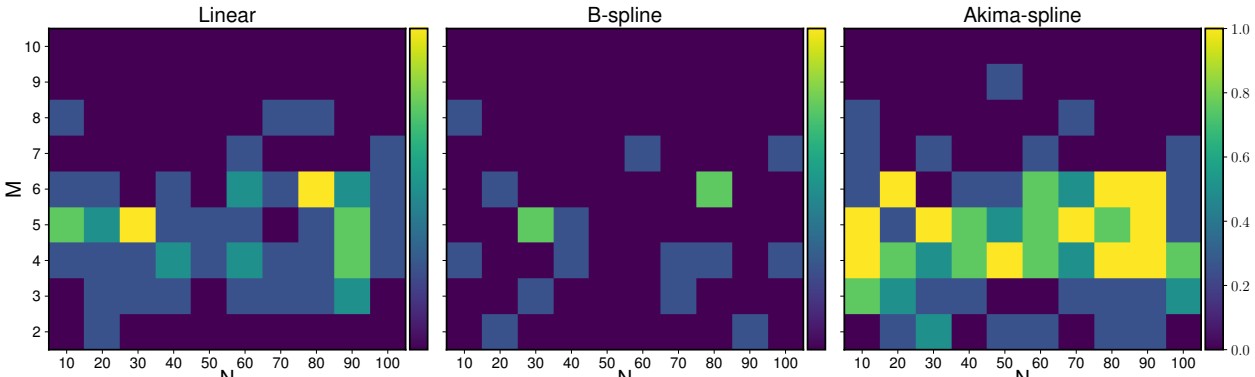

Figure 8: Sweep $M, N$ on `Navigation` environment with $B = 256$ to investigate the interplay between number of batch sample $B$, number of layer $M$, and number of control-waypoints per layer $N$. Each data point is the success rate over 4 seeds. The environment setting is as in Appendix A.3.

$M$ in tensor sampling. Roughly, for all MTP variants, increasing $M$ initially improves task performance, as more layers provide sufficient path complexity, allowing the planner to escape local minima and generate diverse, globally exploratory trajectories. However, beyond a certain point, further increasing $M$ degrades performance. This is due to the exponential growth in the number of possible paths $\mathcal{O}(N^M)$, while the rollout budget $B$ remains fixed. As a result, the sampled trajectory density becomes sparse relative to the vast number of paths, reducing effective coverage and leading to diminished exploration and performance. On the other hand, increasing $N$ consistently improves performance by densifying the search at each layer.

However, this comes at a higher computational cost. Therefore, a careful balance must be struck between $M$ and $N$ to maintain real-time control and effective control exploration.

**Planning Performance.** Table 5 benchmarks the performance of our JAX-based implementation by measuring the wall-clock time of the JIT-compiled planning function on `G1-Standup` task. This function includes a single sampling, single trajectory rollout, single cost evaluation, and single parameter update. Note that this benchmark only serves as an exemplary simulated planning function performance with JAX JIT. In practice, we might have multiple search refinements, or multiple parameter updates in the control loops. The task setting is similar to Table 2, but we set the sim step per plan to 1 with $B = 128$. The results show that the initial JIT compilation incurs a significant one-time cost, as expected for MuJoCo XLA pipelines. However, after compilation, the per-step planning rates across MTP, MPPI, PS, and evolutionary baselines are roughly similar with the same batch sample $B$, as also reflected in Table 6, Table 7, and Table 8. The results confirm that the JIT [s] and Planning Time [ms] are algorithm-agnostic, which depends slightly on batch size $B$ (i.e., Planning Time [ms] logarithmically increases with $B$) and on the environment dynamics. All algorithms remain real-time feasible on GPU-accelerated hardware, when implemented with JAX and MuJoCo XLA.

Table 5: JAX implementation benchmark on `G1-Standup`, evaluated with 5 seeds on an Nvidia RTX 3090.

|  | MTP-Bspline | MTP-Akima | PS | MPPI | OpenAI-ES | DE |
|---|---|---|---|---|---|---|
| JIT Time [s] | $76.4 \pm 1.2$ | $74.62 \pm 2.5$ | $72.35 \pm 4.5$ | $73.87 \pm 4.2$ | $69.87 \pm 1.2$ | $73.62 \pm 3.1$ |
| Planning Time [ms] | $2.7 \pm 0.3$ | $2.7 \pm 0.4$ | $3.1 \pm 0.7$ | $2.6 \pm 0.2$ | $2.9 \pm 0.5$ | $3.2 \pm 0.6$ |

Table 6: Planning performance of **MTP-Akima**. Averaged over 5 seeds on an Nvidia RTX 4090.

| Batch Size $B$ | JIT Time [s] | | | | Planning Time [ms] | | | |
|---|---|---|---|---|---|---|---|---|
| | PushT | Crane | Cube-In-Hand | G1-Walk | PushT | Crane | Cube-In-Hand | G1-Walk |
| 64 | 15.8 | 32.5 | 38.2 | 65.3 | $1.7 \pm 0.1$ | $1.8 \pm 0.1$ | $8.1 \pm 0.3$ | $1.4 \pm 0.1$ |
| 128 | 12.7 | 31.5 | 36.6 | 58.5 | $1.6 \pm 0.1$ | $1.9 \pm 0.1$ | $10.2 \pm 0.7$ | $1.5 \pm 0.1$ |
| 256 | 16.3 | 31.4 | 38.8 | 57.1 | $2.0 \pm 0.2$ | $2.0 \pm 0.4$ | $14.7 \pm 0.8$ | $1.5 \pm 0.1$ |

Table 7: Planning performance of **MPPI**. Averaged over 5 seeds on an Nvidia RTX 4090.

| Batch Size $B$ | JIT Time [s] | | | | Planning Time [ms] | | | |
|---|---|---|---|---|---|---|---|---|
| | PushT | Crane | Cube-In-Hand | G1-Walk | PushT | Crane | Cube-In-Hand | G1-Walk |
| 64 | 14.9 | 32.3 | 37.7 | 62.9 | $1.7 \pm 0.1$ | $1.7 \pm 0.1$ | $8.2 \pm 0.3$ | $1.4 \pm 0.1$ |
| 128 | 18.4 | 29.9 | 36.7 | 58.2 | $1.7 \pm 0.1$ | $1.8 \pm 0.1$ | $10.4 \pm 0.5$ | $1.4 \pm 0.1$ |
| 256 | 14.9 | 30.3 | 37.5 | 55.7 | $1.9 \pm 0.1$ | $1.8 \pm 0.1$ | $15.2 \pm 0.9$ | $1.4 \pm 0.1$ |

Table 8: Planning performance of **OpenAI-ES**. Averaged over 5 seeds on an Nvidia RTX 4090.

| Batch Size $B$ | JIT Time [s] | | | | Planning Time [ms] | | | |
|---|---|---|---|---|---|---|---|---|
| | PushT | Crane | Cube-In-Hand | G1-Walk | PushT | Crane | Cube-In-Hand | G1-Walk |
| 64 | 15.1 | 32.1 | 39.2 | 61.5 | $1.7 \pm 0.1$ | $1.8 \pm 0.1$ | $8.2 \pm 0.3$ | $1.6 \pm 0.1$ |
| 128 | 19.0 | 30.1 | 38.4 | 57.9 | $1.7 \pm 0.1$ | $1.9 \pm 0.1$ | $10.3 \pm 0.4$ | $1.5 \pm 0.1$ |
| 256 | 16.1 | 30.9 | 42.3 | 55.1 | $2.0 \pm 0.1$ | $1.8 \pm 0.1$ | $14.9 \pm 0.7$ | $1.5 \pm 0.1$ |

This demonstrates that MTP, despite its global exploration capabilities, remains suitable for real-time control applications. Our implementation benefits from efficient JIT and `vmap` vectorization in JAX and is compatible with MuJoCo's XLA backend. These design choices ensure that sampling, rollout, and learning components of MTP are fully optimized and scalable, and they support advanced techniques such as online domain randomization. Overall, the benchmark confirms the practicality of deploying MTP in high-performance robotic control loops.

**Softmax Update Effect.** Fig. 9 illustrates the impact of applying `softmax` weighting on elite candidates for updating the mean and standard deviation of control trajectories. The left plot demonstrates smoother and lower-variance control updates over time compared to updates without `softmax` weighting shown on the right. The smooth and stable updates afforded by `softmax` weighting are essential when effectively mixing global and local samples, highlighting its critical role in the MTP performance.

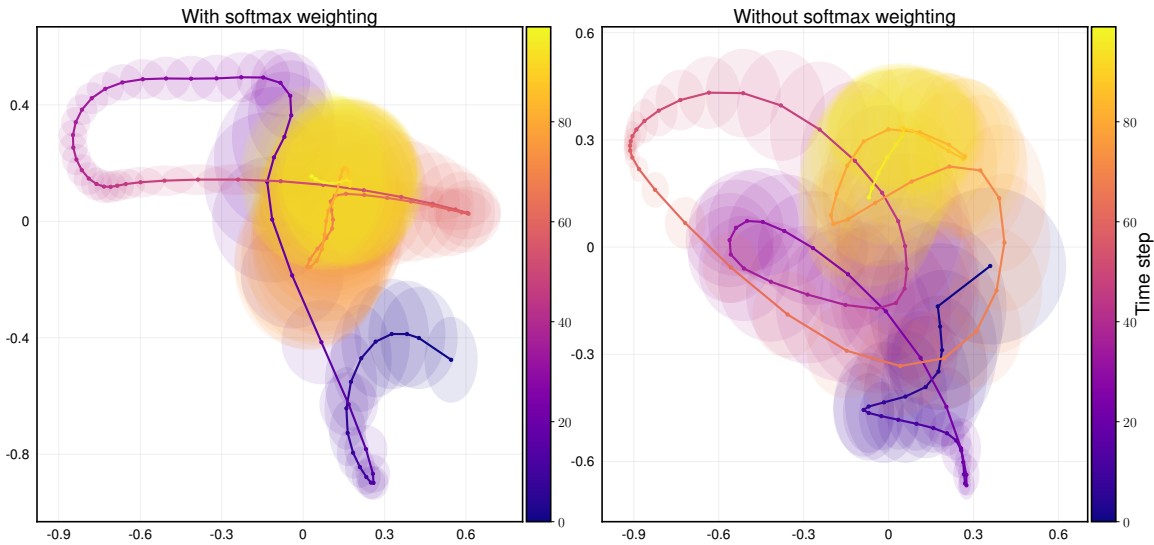

Figure 9: `Softmax` weighting ablation on `PushT` environment. Both control update trajectories converge to near-zero means with large variance at 100 timesteps, signifying task completion.

