# OpenReview forum: "Model Tensor Planning"
_TMLR — Accepted by TMLR_

### Review · Reviewer_fhfo · 2025-05-28

**Summary Of Contributions:**

Sampling-based model predictive control (MPC) often suffers from limited exploration due to locally greedy sampling strategies. This paper aims to explicitly improve exploration in MPC. The proposed method generates a set of control sequences ${U^{k}}_{k=1}^B$ that combine Gaussian samples (from $\mathcal{N}(\mu, \sigma^2)$) with highly exploratory paths generated from tensor sampling, where the portion of them is determined by the mixing parameter $\beta\in(0,1)$. The tensor sampling can be interpreted as uniformly sampling from an N-width, M-depth fully connected graph, where exactly one node is selected per layer. Such paths are structured and highly exploratory.

**Audience:**

Yes

**Broader Impact Concerns:**

Not as far as I can see

**Claims And Evidence:**

Yes

**Requested Changes:**

### Main

1. Respond to the questions in the weakness section.

### Minor

1. The notations $u$, $\mathbf{u}$, $\mathbf{U}$, and $u_t$ should be defined more clear - especially on page 2. In Definition 1, I assume $U(\mathbb{U})$ denotes the uniform distribution over $\mathbb{U}$, but this should be stated explicitly, especially given the heavy reuse of "U" throughout the paper.

2. The cost function $c$ in Section 2.1 is used before definition.

3. Should the average in (4) be $E$ or $B$?

4. You reuse the notation $E$ for the number of elite candidates and the edge set. Would be better to make them different.

5. The “path point” $\mathbf{u}(t)$ in between Eq. (6) and (7) is better to call it “waypoint” here for the sake of consistency.

**Strengths And Weaknesses:**

## Strengths

The approach is quite interesting and effectively encourages exploration by mixing in highly exploratory samples. Under certain conditions, these samples can achieve full coverage of the control space. From what I understand, this work offers a promising direction for future work to further refine the trade-off between exploration and exploitation.

## Weaknesses

1. The coverage result looks reasonable. However, its relevance here is questionable, particularly given how far the actual experimental setting is from the conditions $N,M\rightarrow \infty$ (see Table 3).

2. (Question) The level of exploration in your method seems to be tunable with $N$ and $\beta$. Here, $N$ affects how finely the exploration is, and $\beta$ controls how many exploratory trajectories are. While you discuss the sensitivity of $\beta$, how it should be chosen seems to be indecisive. I might have overlooked it, but there’s little mention of how to choose $N$. Is this important, given that you actually have quite different values in Table 3? Also, have you considered using a time-varying/decaying $\beta$? Otherwise, your method always includes exploratory samples. I was just wondering how it affects the convergence or stability of $(\mu, \sigma^2)$.

3. (Question) Can you explain a bit more about the complexity of sampling with/without replacement? It seems to be quite the opposite to me.

---

> ### Author Response · Authors · 2025-06-12
>
> We thank the reviewer for appreciating the conceptual novelty of mixing high-entropy tensor sampling with local policies and its potential for improving exploration.
>
> ---
> > The coverage result looks reasonable. However, its relevance here is questionable, particularly given how far the actual experimental setting is from the conditions $M, N \to \infty$ (see Table 3).
>
> We acknowledge this and now explicitly discuss the practical settings in Section 3.2 that our theory serves as a guiding principle, while practical settings trade off entropy with success rate and runtime feasibility. Our ablation in Figure 8 demonstrates diminishing returns beyond moderate $M, N$ (w.r.t. fixed sample size $B$) validating this trade-off.
>
> ---
> > (Question) The level of exploration in your method seems to be tunable with $N$ and $\beta$. Here, $N$ affects how finely the exploration is, and $\beta$ controls how many exploratory trajectories are. While you discuss the sensitivity of $\beta$, how it should be chosen seems to be indecisive. I might have overlooked it, but there’s little mention of how to choose $N$. Is this important, given that you actually have quite different values in Table 3? Also, have you considered using a time-varying/decaying $\beta$? Otherwise, your method always includes exploratory samples. I was just wondering how it affects the convergence or stability of $(\mu, \sigma^2)$.
>
> Thank you for your thoughtful question regarding the roles and sensitivities of the hyperparameters $N$ and $\beta$ in our proposed method.
>
> Regarding $N$ (number of nodes per layer in the tensor graph), we agree it directly affects the granularity of the control graph and the diversity of control sequences that can be constructed. In principle, increasing $N$ should enable finer-grained exploration over the trajectory space. However, as illustrated in Figure 8, we observe diminishing returns when $N$ increases while keeping the total number of sampled trajectories $B$ fixed. This is because although the underlying graph becomes denser, the number of explored paths remains constant, resulting in only marginal performance improvements. Therefore, we recommend choosing $N$ proportionally to $B$ and within the bounds of available GPU memory, in order to maintain computational efficiency without oversampling from a small sample size.
>
> For $\beta$, which determines the ratio between exploratory (tensor-sampled) and exploitative (nominal sampling) samples, we acknowledge its delicate nature. As shown in our ablation, high $\beta$ values introduce strong exploration, which may benefit tasks with sparse rewards or challenging local minima (e.g., PushT, Cube-in-hand, Navigation). Conversely, in tasks requiring high stability or precise actuation, such as G1-Standup or G1-Walk, lower $\beta$ values (e.g., 0.05–0.1) tend to yield better performance by favoring consistent behavior while still injecting enough exploration to escape suboptimal solutions. We clarify these points in Section 3.2 and Appendix A.4 to better guide readers on practical hyperparameter choices and trade-offs.
>
> We appreciate your suggestion on using a time-varying or decaying $\beta$. We did not explore this explicitly in our current experiments due to implementation difficulty with JAX static tensor shape requirement (see https://docs.jax.dev/en/latest/jit-compilation.html#why-can-t-we-just-jit-everything). The reason is that changing $\beta$ per iteration will change the tensor shapes of both local/tensor sampling in the loop (cf. Line 4-8, Algorithm 1), which induces JIT-ing impossible. However, we agree it presents a promising direction. A decaying $\beta$ schedule would allow the controller to begin with high entropy for global exploration and gradually shift toward more focused, exploitative behavior as optimization converges. This could improve convergence stability and better reflect the evolving uncertainty in $(\mu, \sigma^2)$. We plan to investigate this idea in future work, potentially drawing parallels to temperature annealing strategies used in other probabilistic control settings. Thank you again for raising this valuable question.
>
> ---
> > (Question) Can you explain a bit more about the complexity of sampling with/without replacement? It seems to be quite the opposite to me.
>
> We added clarifications in Section 3.1. Sampling without replacement adds overheads due to re-indexing or tree-traversing to get batch of sequence indices (depending on low-level implementation of sampling) but offers better diversity. In practice, we use sampling with replacement, which does not really affect diversity (cf. Figure 1 last row), is faster and scales well with JAX vectorized operations.
>
> ---
> > Minor: unclear notations and inconsistent symbols.
>
> Thank you. We corrected notation inconsistencies, explicitly defined all reused symbols in the revised manuscript, and renamed "path point" to "waypoint" for consistency.

---

### Review · Reviewer_2keG · 2025-06-03

**Summary Of Contributions:**

This paper introduces Model Tensor Planning (MTP), a tensorized sampling-based planning method designed to address the mode-collapse problem commonly encountered in classical methods, such as the Cross Entropy Method (CEM). The core idea is to discretize the planning space into a graph with M layers of N nodes each, where M corresponds to the planning horizon and N is the number of bins to discretize the control space. To generate a trajectory, one can sample a node from each layer and interpolate them using B-spline or Alkema-spline. The planning method maintains a catergorical distribution at each layer which is iteratively refined by refitting to the elite samples, similar to CEM. The paper proves that this tensorized planning method covers the full planning space as M and N approach infinity (achieving maximal entropy). Empirically, they show the effectiveness of MTP across a suite of simulated benchmarks, including Push-T, Crane, Hand-In-Cube, Walker, and Humanoid-Stand, where MTP demonstrates improved exploration behavior and faster convergence compared to continuous-space planning methods.

**Audience:**

Yes

**Broader Impact Concerns:**

The paper broadly applies to robotics problems. Despite evaluating purely in simulated environments, the potential of real-world applications warrants a Broader Impact Statement. I would kindly request the authors to add a Broader Impact Statement on the safety concerns of applying the method to real robots.

**Claims And Evidence:**

Yes

**Requested Changes:**

The paper is well-written and easy to follow. All the claims are supported by evidence. Therefore, I do not have requested changes.

**Strengths And Weaknesses:**

**Strengths**
1. MTP offers a simple and practical solution to the mode collapse problem in classical sampling-based planning methods, where the candidate distributions are often represented by Gaussian distributions. By converting the planning space in to an M-layer graph and maintaining a categorical distribution at each layer, MTP can plan over expressive, high-entropy distributions of trajectories without sacrificing optimality.
2. MTP can be efficiently implemented using accelerated tensor libraries (e.g. JAX), which combined with accelerated simulations (e.g. BRAX) offer a significant speedup over classical methods.
3. The paper performs thorough hyperparameter sweeps and ablation studies to help understand the contribution of each component.

**Weaknesses**
1. Despite theoretically converging to uniform coverage over the planning space in the limits, in practice the hyperparameters M and N are chosen to trade off fidelity and computation complexity.

---

> ### Author Response · Authors · 2025-06-12
>
> We are grateful for the reviewer's recognition of the strengths of MTP in addressing mode collapse, tensorization, and implementation efficiency.
>
> > The paper broadly applies to robotics problems. Despite evaluating purely in simulated environments, the potential of real-world applications warrants a Broader Impact Statement. I would kindly request the authors to add a Broader Impact Statement on the safety concerns of applying the method to real robots.
>
> Thank you for the helpful suggestion. We have now included a dedicated Broader Impact section at the end of the paper. It discusses the potential of MTP to contribute a new direction in efficient, high-entropy control generation for robotics. By introducing a structured sampling mechanism that supports real-time, differentiable planning, MTP opens the door to more robust decision-making in high-dimensional, contact-rich environments. At the same time, we acknowledge safety-related considerations: while MTP operates within bounded control limits, its rapid control changes may require actuators and control stacks to be carefully tuned to handle high-frequency updates in practice.
>
> We also address the integration of MTP into real-world systems, noting that — like other MPC-based methods — it requires accurate robot and environment state estimates, typically via a state estimator. These states serve as the starting point for MJX-based parallel rollouts (digital twin simulation). We view this as an engineering challenge centered around ensuring low-latency communication between the physical robot and its digital counterpart. With proper safeguards, we believe MTP offers a promising path toward scalable and safe real-world deployment.

---

### Review · Reviewer_BBca · 2025-06-04

**Summary Of Contributions:**

The authors propose Model Tensor Planning (MTP), to effectively balance exploration and exploitation in sampling-based MPC. Authors first reformulate the sampling as tensor operations over randomized multipartite graphs to generate diverse candidates and introduce $\beta$-mixing mechansim for exploration-exploitation trade-off. Authors conduct several experiments on high-dimensional real-time control tasks.

**Audience:**

Yes

**Claims And Evidence:**

Yes

**Requested Changes:**

I have several minor comments as follows:

- Experiment results of Walker: It seems that there is no big difference between baselines and MTP variants in Walker task. While authors say it is due to absence of applying modeling error, more detailed discussion may required.

- Experiments results of G1-Standup and G1-walk: Those benchmarks represent high-dimensional action spaces, which requires the capability of balancing exploration and exploitation systemically. However, it seems that there is no big difference in terms of performance between MPPI and MTP variants. It is quite strange as MPPI should suffer from balancing exploration-exploitation in high-dimensional spaces. It would be nice to interpret these observations clearly.

- In my humble opinion, the implementation with JIT and vmap vectorization in JAX is one of the key contributions of the paper. However, i can find the time complexity analysis in the appendix with a single table. I recommend to compare wall clock time of algorithms across different benchmarks and also analyze the effect of B in terms of runtime, which might be interesting.

**Strengths And Weaknesses:**

Strenghts)
- Clearly point out the limitation of prior sampling-based MPC algorithms with motivating experiments.
- Clearly demonstrate the robustness of the proposed method.

Weakness)
- Insufficient explanation on experiment results.
- Lack of analysis on time-complexity.

---

> ### Author Response · Authors · 2025-06-12
>
> We thank the reviewer for highlighting the clarity of our motivation and the robustness of our method.
>
> ---
> > Experiment results of Walker: It seems that there is no big difference between baselines and MTP variants in Walker task. While authors say it is due to absence of applying modeling error, more detailed discussion may required.
>
> We agree and now clarify this in the text. The Walker task exhibits a rather simple dynamics model and is less sensitive to the sampling distribution due to its relatively simple contact model. This serves as a sanity check to see if MTP and classical MPPI/PS perform similarly in simple cases. We emphasize this in the updated experimental discussion (Section 4.2).
>
> ---
> > Experiments results of G1-Standup and G1-walk: Those benchmarks represent high-dimensional action spaces, which requires the capability of balancing exploration and exploitation systemically. However, it seems that there is no big difference in terms of performance between MPPI and MTP variants. It is quite strange as MPPI should suffer from balancing exploration-exploitation in high-dimensional spaces. It would be nice to interpret these observations clearly.
>
> We apologize for not pointing out clearly. We elaborate that although MPPI performs similarly well to MTP variants in G1-Standup in terms of standing up rate. However, MPPI is more unstable under noise after standing up (see the MPPI curve after the red dashed line in Fig. 4, especially under model errors with domain randomization setting).
>
> For G1-Walk, MPPI and PS curves diverges to motion capture trajectory very early, hinting the unstable walking behavior. MTP variants are able to keep the walk posture longer (see visualization here: https://sites.google.com/view/tensor-sampling/).
>
> We would like to highlight a broader insight into the exploration–exploitation trade-off in sampling-based MPC methods. In tasks with well-shaped or dense reward structures and moderately nonlinear dynamics, exploration becomes less critical and nominal sampling methods (e.g., MPPI, PS) often suffice—explaining MPPI’s strong performance in \texttt{G1-Standup}, which benefits from fully actuated dynamics and informative rewards. However, in more challenging scenarios involving sparse rewards or highly nonlinear dynamics, such as \texttt{PushT} and under domain shifts in \texttt{Crane}, these locally guided strategies tend to struggle with inadequate exploration, often converging to suboptimal solutions or showing high performance variance. A representative case is the \texttt{Navigation} task (Figure 3), where the agent must discover velocity sequences to bypass obstacles and reach the goal — a setting in which local Gaussian sampling clearly fails by getting trapped in local minima. To address these challenges, MTP introduces a structured high-entropy sampling mechanism along with a simple yet effective $\beta$-mixing strategy that balances global exploration and local exploitation. With careful tuning of $\beta$, $M$, and $N$, MTP demonstrates robust and consistent performance across diverse tasks. We have clarified these points further in Appendix A.4.
>
> ---
> > In my humble opinion, the implementation with JIT and vmap vectorization in JAX is one of the key contributions of the paper. However, i can find the time complexity analysis in the appendix with a single table. I recommend to compare wall clock time of algorithms across different benchmarks and also analyze the effect of B in terms of runtime, which might be interesting.
>
> We have now added runtime Table 6, 7, 8 in Appendix A.5 measuring JIT [s] and Planning Time [ms] of MTP-Akima, MPPI, and OpenAI-ES across varying representative batch sizes and environments. The results confirm that the JIT and Planning Time are algorithm-agnostic, which depends slightly on batch size $B$ (i.e., Planning Time logarithmically increases with $B$) and on the environment dynamics.
> For instance, in contact-rich tasks such as the Cube-In-Hand, solving contact dynamics results in higher simulation times, which are reflected in higher planning times.
> All algorithms remain real-time feasible on GPU-accelerated hardware, when implemented with JAX and MuJoCo XLA.
> The major bottleneck is be the GPU memory, which caps the amount of parallel samples one can evaluate.

---

### Author Response · Authors · 2025-06-12
**Revision Note**

We sincerely thank the reviewers for their thoughtful and constructive feedback. We appreciate the time and effort spent evaluating our submission, Model Tensor Planning (MTP). We are happy that reviewers found the proposed method both well-motivated and promising, especially in addressing the exploration limitations of sampling-based MPC. We are grateful for the recognition of our theoretical insights, practical implementation with JAX, and extensive evaluations.

We have uploaded the revised manuscript, where we:

- Clarified experiment results, particularly for sanity-check Walker task and high-dim tasks like Humanoid in Sec. 4.2.
- Added runtime wall-clock comparisons and discussion on scalability with respect to batch size in Appendix A.5.
- Provided guidance on the choice of key hyperparameters (especially $\beta$ and $N$) in Sec. 3.2 and Appendix A.4
- Expanded notation clarity and improved consistency in definitions and terminology.
- Included a Broader Impact Statement discussing implications for real-world robotics.

Below we provide point-by-point responses to each reviewer.

---

### Decision · Action_Editor_eMBQ · 2025-07-14

**Recommendation:** Accept as is

**Audience:**

Yes

**Audience Explanation:**

The work addresses a fundamental problem in sampling-based model predictive control, the mode collapse and poor exploration of locally greedy sampling schemes. This is a practically relevant issue for robotics researchers working on contact-rich and nonlinear control tasks. The efficient JAX implementation and compatibility with accelerated simulation environments adds practical value for the community.

**Claims And Evidence:**

Yes

**Claims Explanation:**

All three reviewers acknowledged that the paper provides clear evidence for its claims through extensive experiments across diverse robotic tasks. The theoretical results on asymptotic path coverage and maximum entropy are sound, and the empirical evaluation demonstrates MTP's effectiveness in addressing exploration limitations of sampling-based MPC. The authors adequately addressed reviewer concerns about experimental interpretations, runtime analysis, and hyperparameter guidance.